# The importance of model horizontal resolution on simulated precipitation in Europe – from global to regional models

Gustav Strandberg[1,2], Petter Lind[1,2,3]

[1]Rossby Centre, Swedish Meteorological and Hydrological Institute, SMHI, Norrköping, SE-602 19, Sweden
[2]Bolin Centre for climate research, Stockholm University, Stockholm, SE-106 91, Sweden
[3]Department of meteorology, Stockholm University, Stockholm, SE-106 19, Sweden

*Correspondence to*: Gustav Strandberg (gustav.strandberg@smhi.se)

**Abstract.** Precipitation is a key climate variable that affects large parts of society, especially in situations with excess amounts. Climate change projections show an intensified hydrological cycle through changes in intensity, frequency, and duration of precipitation events. Still, due to the complexity of precipitation processes and their large variability in time and space, climate models struggle to represent precipitation accurately. This study investigates the simulated precipitation in Europe in available climate model ensembles that cover a range of model horizontal resolutions. The ensembles used are: Global climate models (GCMs) from CMIP5 and CMIP6 (~100-300 km horizontal grid spacing at mid-latitudes ), GCMs from the PRIMAVERA project at sparse (~80-160 km) and dense (~25-50 km) grid spacing and CORDEX regional climate models (RCMs) at sparse (~50 km) and dense (~12.5 km) grid spacing. The aim is to seasonally and regionally over Europe investigate the differences between models and model ensembles in the representation of the precipitation distribution in its entirety and through analysis of selected standard precipitation indices. In addition, the model ensemble performances are compared to gridded observations from E-OBS.

The impact of model resolution on simulated precipitation is evident. Overall, in all seasons and regions the largest differences between resolutions are seen for moderate and high precipitation rates, where the largest precipitation rates are seen in the RCMs with highest resolution (i.e. CORDEX 12.5 km) and smallest in the CMIP GCMs. However, when compared to E-OBS the high-resolution models most often overestimate high-intensity precipitation amounts, especially the CORDEX 12.5 km resolution models. An additional comparison to a regional data set of high-quality lends, on the other hand, more

confidence to the high-resolution model results. The effect of resolution is larger for precipitation indices describing heavy precipitation (e.g. maximum one-day precipitation) than for indices describing the large-scale atmospheric circulation (e.g. the number of precipitation days), especially in regions with complex topography and in summer when precipitation is predominantly caused by convective processes. Importantly, the systematic differences between low resolution and high resolution remain also when all data are regridded to common grids of 0.5°×0.5° and 2°×2° prior to analysis. This shows that the differences are effects of model physics and better resolved surface properties and not due to the different grids on which the analysis is performed. PRIMAVERA high resolution and CORDEX low resolution give similar results as they are of similar resolution.

Within the PRIMAVERA and CORDEX ensembles there are clear differences between the low- and high-resolution simulations. Once reaching ~50 km the difference between different models is often larger than between the low- and high-resolution versions of the same model. For indices describing precipitation days and heavy precipitation the difference between two models can be twice as large as the difference between two resolutions, in both the PRIMAVERA and CORDEX ensembles. Even though increasing resolution improves the simulated precipitation in comparison to observations, the inter-model variability is still large, particularly in summer when smaller scale processes and inter-actions are more prevalent and model formulations (such as convective parameterizations) become more important. .

# 1 Introduction

Precipitation is a key climate variable affecting the environment and human society in different ways and on several temporal and spatial scales. In particular, heavy precipitation events may lead to large damages caused by floods or landslides, while the absence of precipitation may cause droughts and has impact on water- and hydropower supply. In recent decades there has therefore been extensive study, and considerable advancement in our understanding, of the response of extreme precipitation to climate change (O'Gorman, 2012; Kharin et al. 2013; Donat et al., 2016; Pfahl et al. 2017). For example, it is widely held through theoretical considerations and model experiments that extremes will respond

differently than changes in mean precipitation (e.g. Allen and Ingram 2002; Pall et al 2007; Ban et al.,
55  2015).


Still, the simulation of precipitation in weather and climate models is challenging because of the wide
range of processes involved that acts and interacts on widely different temporal and spatial scales.  An
accurate representation of precipitation in models requires skill in simulating (1) the large-scale
circulation, (2) interaction of the flow with the surface, and, (3) convection and cloud processes. With
the typical horizontal grid resolution of O (100 km) of global climate models (GCMs) point (1) can to a
large extent be properly represented but less so for (2) and (3) (e.g. van Haren et al., 2015; Champion et
al., 2011; Zappa et al., 2013). In particular, atmospheric convective processes are not resolved and
needs to be treated with convection parameterizations. As the range of scales resolved is broadened
through refining the horizontal grid spacing the simulation of precipitation generally improves. This is
achieved through more realistic representation of surface characteristics (such as topography, coastlines
and inland lakes and water bodies) and through more accurately solving the motion equations resulting
in more accurate horizontal moisture transport and moisture convergence (Giorgi and Marinucci 1996;
Gao et al. 2006; Prein et al. 2013a). Indeed, GCMs with ~25-50 km grid spacing show promise to
improve simulation of precipitation (van Haren et al., 2015; Delworth et al., 2012; Kinter et al., 2013;
Haarsma et al., 2016; Roberts et al., 2018a; Baker et al., 2019).

Dynamical down-scaling of GCMs with regional climate models (RCMs) allows for even finer grids
which leads to more detailed information of and further improvements in regional and local climate
features, for example spatial patterns and distributions of precipitation in areas of complex terrain
(Rauscher et al., 2010; Di Luca et al., 2011; Prein et al., 2013b). This can also have important
implications for climate change signals. Giorgi et al. (2016) found that an ensemble of RCMs at ~12 km
grid spacing showed consistently an increase in summer precipitation over the Alps region which
contrasted to the forcing GCMs that instead showed a decrease. The different responses were attributed
to increased convective rainfall in the RCMs due to enhanced potential instability by surface heating
and moistening at high altitudes not captured by the GCMs. Differences in the treatment of aerosols are
also identified as a reason for differences in climate response between RCMs and GCMs (Boé et al.,
2020; Gutiérrez et al., 2020). RCMs are constrained by the lateral boundary conditions provided by the
forcing GCM and studies of RCM ensembles have shown that the choice of forcing GCM have
introduced the major part of the overall uncertainty in regional climate (e.g. Déqué et al., 2007;
Kjellström et al., 2011). This effect is relatively more important for large-scale precipitation systems,
for example frontal systems associated with extra-tropical cyclones. In seasons and regions when
smaller scale processes like convection dominate, for example in summer over mid-latitudes, simulated
precipitation is to a larger degree dependent of the RCM itself, in terms of grid resolution and sub-grid
scale parameterizations (e.g. Iorio et al., 2004). A recent study investigated the effects of model
resolution on local precipitation on short time scales and found that the 12.5 km simulations better
represent daily and sub-daily extreme and mean precipitation, also when simulations are aggregated to
50 km (Prein et al., 2016). They note, however, that the results are highly dependent on which
observations the simulations are compared with, and that improvements are seen for the ensemble
mean, and not necessarily for each individual model. In similar studies as the present one Iles et al.
(2019) and Demory et al. (2020) compare simulations from the CORDEX, CMIP5 and PRIMAVERA
ensembles. The results show increases  in precipitation with resolution and , when compared to a
mixture of E-OBS and high spatial-resolution gridded national datasets, CMIP5 underestimates
precipitation amounts while CORDEX overestimates it,  the effect of grid resolution  being largest in
areas with complex topography. They also find that PRIMAVERA performs similarly to CORDEX
when run on the same resolution, which is interesting regarding that the PRIMAVERA models are
developed for low resolutions. Iles et al. (2019) concluded from the considerable inter-model
differences that improvements are seen for the ensemble mean rather than for individual models.
Although increased grid resolution often leads to improved simulation of precipitation, convection is
usually not resolved by the model dynamics, even at grid spacings of around 10 km, but is instead
parameterized (although it might be possible to turn off the parameterization already at this kind of
resolution (Vergara-Temprado et al., 2019)). The choice of convection parameterization can have
various effects on the occurrence and amount as well as on the onset timing and location (e.g. Dai et al.,
1999; Dai 2006; Stratton and Stirling, 2012; Gao et al., 2017). Commonly, models with parameterized
convection exhibit biases in the diurnal precipitation cycle (Liang, 2004; Brockhaus et al., 2008; Gao et
al. 2017), sometimes regardless of increases in grid resolution (Dirmeyer et al., 2012). In addition,
models of coarse resolution often suffer from simulating precipitation over too large area compared to
observations, and usually also too many days with weak precipitation (the "drizzle" problem) (e.g. Dai,
2006, Stephens et al., 2010). At sufficiently high resolution (< 4 km) models start to largely resolve
deep convection enabling the parameterization to be turned off, so called "convection-permitting"
models (Prein et al., 2015; Vergada-Temprado et al., 2019). Convection-permitting regional climate
models (CPRCMs) are widely shown to reduce, at least to some extent, these biases, most evidently by
improving the match of the diurnal cycle to observations (e.g. Prein et al., 2013a; Ban et al., 2014;
Brisson et al., 2016; Gao et al., 2017; Leutwyler et al., 2017; Belušić et al. 2020) and better
representation of sub-daily high-intensity precipitation events (e.g. Ban et al., 2014; Kendon et al.,
2014; Fosser et al., 2015; Lind et al., 2020) than models with parameterized convection. A major draw-
back using these high-resolution climate models is the very high computational cost, making their use in
ensembles to only recently emerge (Coppola et al., 2018).

The aim of this study is to:
i. Investigate to what extent a large number of global and regional climate models can reproduce
observed daily precipitation climatologies and characteristics over Europe.
ii. Investigate how model horizontal grid resolution in either global or regional models affect the
simulated precipitation in Europe; are there systematic differences and if so, are these persistent for
different parts of Europe and for different seasons.

To this end, GCMs of standard resolution from the CMIP5 (Climate Model Intercomparison Project
phase 5, Taylor et al., 2012) are compared with GCMs which participated in the HighResMIP (High
Resolution Model Intercomparison Project, Haarsma et al., 2016) experiment within the H2020-EU-
project PRIMAVERA. These models are: ECMWF-IFS (Roberts et al., 2018b), HadGEM3-GC31
(Roberts et al., 2019), MPI-ESM1.2 (Gutjahr et al., 2019), CNRM-CM6.1 (Voldoire et al., 2019) and
EC-Earth3P (Haarsma et al., 2020). Furthermore, the first results from the CMIP6 (Climate Model
Intercomparison Project phase 6, Eyring et al., 2016) GCMs are included in the analysis. The GCMs are
compared with RCMs from CORDEX (COordinated Regional Downscaling EXperiment, Gutowski et
al., 2016). This allows for comparisons of different generations of models, global versus regional
models and the impact of model horizontal grid resolutions. For a few cases, the same model version
has been applied at two different grid resolutions which allows for investigating the impact of resolution
alone. The simulated daily precipitation is analysed both in terms of precipitation intensity distributions
and through a collection of standard precipitation-based indices.

## 2 Models and Methods


### 2.1 Global and regional models


The models used in this study are a selection of CMIP5 global models (corresponding to ~100-300 km
horizontal grid spacing at mid-latitudes); the high (~25-50 km) and low (~80-160 km) resolution
versions of the PRIMAVERA global models and the first available runs from CMIP6 (~100-300 km);
and finally, a selection of CORDEX RCMs (at 12.5 and 50 km mid-latitude grid spacing). The low-
resolution versions in each model ensemble is called LR, and the high-resolution HR. Note that not the
full CMIP5, CMIP6 and CORDEX ensembles are used, but rather "ensembles of opportunity" for
which daily precipitation were readily available. Table 1 lists the GCM ensembles used. Table 2 lists
the GCM RCM combinations used in the CORDEX ensembles. The simulated precipitation for all
models is analysed over the PRUDENCE regions in Europe (Fig. 1; Christensen & Christensen, 2007).
Prior to analysis all grid points over sea are filtered out, and then for each region and model we
calculate precipitation characteristics for all remaining land grid points. The simulations are analysed on
their native grids, because this is the kind of data that users of climate simulations will face, and since
all interpolation may alter precipitation characteristics (Klingaman et al., 2017). Nevertheless, to
investigate all aspects of changed resolution it is sometime necessary to compare simulations on a
common grid. In these cases, the results are also aggregated to two common grids with 2°×2° and
0.5°×0.5° grid spacing respectively.

**2.2 Observations**

Climate model evaluation exercises often rely, when possible, on gridded reference data sets. In this study daily precipitation sums in models are compared with data from E-OBS version 19.0e at 0.1° and 0.25° grid spacing (Cornes et al., 2018). E-OBS comprise daily station values interpolated onto a grid that spans the entire European continent. The main advantage of using E-OBS is the large geographical coverage at a relatively high resolution available over an extended (climatological) time period. It enables a consistent model-observation comparison over the whole continental part of Europe, with its varying climatological and environmental characteristics.

Gridded products, such as E-OBS, involves spatial analysis and interpolation of point measurements onto a regular grid, and are inherently associated with uncertainties originating from both non-climatic influences (e.g. inaccuracies in measurement devices or relocation of measurement sites) and from sampling issues associated with weather and environmental conditions, for example in situations with snowfall in windy conditions (Kotlarski et al. 2019; Rasmussen et al., 2012). The quality of such data sets largely depends on the availability of stations to base the interpolation on, implying that in regions where station density is low the quality of the gridded product is also lower (Herrera et al. 2019). For precipitation this is of even greater importance due to its highly heterogeneous character in both time and space, in particular for high-intensity precipitation events (extremes). These are often local in character (temporally and spatially), even in cases when embedded in larger (synoptic) scale precipitation systems, and can thus be heavily undersampled (Herrera et al. 2019; Prein and Gobiet 2017). Furthermore, mountainous areas act as strong forcing of precipitation giving rise to large spatial variability over the terrain. Combined with the lack of dense networks of stations in these regions, and usually also a higher occurrence of snowfall, makes it very difficult to achieve highly reliable data over mountains (e.g. Hughes et al. 2017; Lundquist et al. 2019).

The quality of E-OBS varies over Europe (see Fig. 1 in Cornes et al. 2018); the station density is for example very high over Scandinavia, Germany and Poland, while it is lower in Eastern Europe and in the Mediterranean region. Gridded regional or national data sets may offer higher quality as these are

generally based on a denser station network and are often also provided with higher spatial and/or
temporal resolution compared to E-OBS (Kotlarski et al. 2019, Prein and Gobiet 2017). Here, we limit
the comparison to E-OBS only. However, to assess the impact of high-quality regional data, an
additional analysis of the precipitation distributions was performed, using ASoP analysis (see Sec. 2.3),
comparing models and E-OBS against the NGCD (Nordic Gridded Climate Dataset, Lussana et al.
2018) data set. NGCD is based on daily station data for precipitation and temperature, interpolated onto
a 1x1 km grid covering Scandinavia.

**2.3 ASoP and precipitation indices**
To investigate the effect of model grid resolution on the full distributions of daily precipitation
intensities, we use the ASoP (Analysing Scales of Precipitation) method (Klingaman et al., 2017;
Berthou et al., 2018). ASoP involves splitting precipitation distributions into bins of different intensities
and then provides information of the contributions from each precipitation intensity separately to the
total mean precipitation rate (i.e. given by all intensities taken together). In the first step, precipitation
intensities are binned in such a way that each bin contains a similar number of events, with the
exception of the most intense events, which are rare. The actual contribution (in mm) of each bin to the
total mean precipitation rate is obtained by multiplying the frequency of events by the mean
precipitation rate. The sum of the actual contributions from all bins gives the total mean precipitation
rate. The fractional contribution (in %) of each bin is further obtained by dividing the actual
contributions by the mean precipitation rate. In this case, the sum of all fractional contributions is equal
to one, thus the information provided by fractional contributions is predominantly about the shape of the
distribution. Taking the absolute differences between two fractional distributions and sum over all bins
gives a measure of the difference in the shapes of the precipitation distributions. This is here called the
"Index of fractional contributions". Since E-OBS precipitation intensities, in contrast to model data, are
not continuous, the resulting ASoP factors for E-OBS tend to be noisy, especially for lower intensities.
In order to facilitate the interpretation of the results, the regionally averaged ASoP factors for E-OBS
were smoothed to some extent by using a simple filter.

The ASoP method is here applied to grid points pooled over target regions (Fig. 1) separately and the result is a distribution for each model showing the probability of different precipitation intensities based on daily precipitation. Most results presented here concern the actual contributions, both to limit the number of figures and because these factors conveniently provide information on both shape of distributions as well as the mean values. The ASoP distributions of all analysed models are used to compare model behaviour and performance. In particular to see how changing the grid resolution affects different parts of the distribution, for example if contributions from low and high precipitation intensities are different.

In addition to ASoP, a number of indices based on daily precipitation (listed in Table 3) are calculated for the same regions. For each model, the indices are calculated separately for each grid point within a region (land points only), and the values are then pooled to calculate percentiles representing the region. This also means that the calculated model spread reflects geographical and not temporal variability. The index percentiles are represented by box plots (Sect. 3).

# 3 Results

## 3.1 ASoP analysis

### 3.1.1 Annual precipitation

Since the ASoP results are very similar between CMIP5 and CMIP6 GCMs (not shown), the results presented here include only one of these ensembles, CMIP6. Figure 2 presents the actual contributions (normalized bin frequency × mean bin rate) for annual daily precipitation over four of the PRUDENCE regions: Scandinavia, mid-Europe, the Alps and the Mediterranean. In general, the model ensembles have higher amounts of precipitation compared to E-OBS, signified by larger contributions at low (< 2-3 mm day$^{-1}$) and moderate-to-high (> 5-10 mm day$^{-1}$) intensities. An exception is the CMIP6 ensemble that instead shows lower contributions for moderate-to-high precipitation intensities, i.e. above 10-20 mm day$^{-1}$ (Scandinavia, mid-Europe and the Alps) or between 5-20 mm day$^{-1}$ (Mediterranean). CMIP6 also tends to have the largest overestimates of contributions from the lower intensities (below 5 mm

day$^{-1}$). Another consistent feature is that the probabilities for the higher intensities (above 15 mm day$^{-1}$)
increase with increasing grid resolutions of respective model ensemble, and consequently the
contributions become increasingly larger than E-OBS (Fig. 2). This is most evident for the Alps region
where the CMIP6 models (100-300 km grid spacing) clearly give smaller contributions than E-OBS and
the PRIMAVERA models (25-160 km), the latter having smaller contributions than the CORDEX LR
models (50 km) and the CORDEX HR models (12.5 km). The higher resolution models peak at higher
intensities and have wider distributions with larger contributions from high-intensity daily rates. The
sensitivity of model grid resolution to precipitation amounts and variability in association with areas
with complex and steep topography (e.g. Prein et al., 2015) is most likely the main reason for the large
differences between model ensembles in the Alps region. For example, the upper end of the CMIP6
distributions is around 50 mm day$^{-1}$ while corresponding part in CORDEX HR models is around 100
mm day$^{-1}$ (bottom right panel in Fig. 2). To further verify the results, the same analysis was performed
after all data had been interpolated (conservatively) to two common grids; one at 2°×2° resolution and
one at 0.5°×0.5° degree resolution (Figs. S1 and S2 in Supplementary). The interpolation to either grid
has an overall small impact on the results. With the coarser grid (2°×2°) the ASoP actual contributions
have relatively larger contributions from the bulk part and a smaller contribution from the highest
intensities, as expected from the smoothing effect of interpolation. These results provide increased
confidence in the conclusions drawn from analysis on native grids.

### 3.1.2 Seasonal precipitation

Further insight can be gained by investigating seasonal differences (Fig. 3).  In winter (DJF) the model
ensemble means generally overestimate total mean precipitation compared to E-OBS (i.e. total areas
under the curves showing differences are positive). The bulk of the distributions are slightly shifted to
higher precipitation rates and also to higher contributions (except for the Mediterranean region). The
largest inter-ensemble differences are seen for the Mediterranean where CORDEX HR shows the
largest shift from E-OBS towards contributions from higher precipitation rates, and PRIMAVERA is
similar to CORDEX LR. In summer (JJA), the ensemble means show larger contributions from
intensities above 10-15 mm/day than E-OBS, especially in CORDEX HR. However, as this is in many

cases compensated by lower contributions from rates between 2-10, the total mean precipitation biases are smaller than in winter. While the CORDEX ensemble means indicate larger total mean precipitation in France and Mediterranean, CMIP6 produces in all regions higher contributions from low-to-moderate ($< \sim 5$ mm/day) compared to E-OBS and lower contributions from higher intensities. Furthermore, there is a tendency in all regions of a larger spread within each model ensemble in JJA than in DJF (see coloured shadings in Fig. 3). Even though it is a very crude estimate of the spreads (the 5-95 percentile range in respective model ensemble), it can be argued that the differences in part is related to the seasonally prevailing weather conditions. In winter the North Atlantic storm track is in its active phase with frequent passings of synoptic weather systems over Europe. These features are generally well represented in climate models – hence larger consistency with associated precipitation across models. In summer, on the other hand, synoptic activity is reduced and convective processes (either as isolated or organized systems or embedded in larger scale features like fronts) become more prominent in precipitation events. Sensitivity to model grid resolution and physics parameterizations (e.g. convection parameterization) is larger during this season. The larger summertime spread in ensembles seen in Fig. 3 might then reflect larger uncertainties associated with model resolution and formulation. It is further noted that the ensemble spread is not increased as much (from winter to summer) over northern/north-western Europe which is relatively more affected by synoptic scale events during summer compared to southern parts of Europe (not shown).

Model ensemble differences for all regions and seasons are summarized in Figure 4, with E-OBS as reference. In spring (MAM) and winter (DJF) all ensembles have higher total mean precipitation in all regions. In summer (JJA) and autumn (SON) biases are also mostly on the positive side but smaller (primarily for GCM ensembles), and in some regions close to zero or slightly negative (e.g. the Alps, East Europe, Iberian Peninsula). Often there is an indication of a positive correlation between differences in mean (x-axis in Fig. 4) and differences in fractional contributions (y-axis, which indicates overall differences in the shape of the distributions), as seen for example in France or Mid-Europe regions. However, there are also cases with large differences in the shape but small total mean precipitation biases, for example the CMIP ensembles in JJA and SON over the Alps, suggesting

compensating effects from different parts of the precipitation distribution. The overall spread is also highly variable between the regions; Scandinavia, Mid- and East-Europe and the British Isles are characterized by relatively smaller inter-ensemble differences, while in the Alps and Mediterranean the spread is large. The spread is in some regions dominated by inter-seasonal differences, e.g. in Mid-Europe and France, where typically the largest differences (in terms of both total means and distribution shapes) occur in DJF and MAM and smaller spreads in JJA and SON. In the Alps, Iberian Peninsula and the Mediterranean regions, however, the relatively larger inter-ensemble differences lead to an increased overall spread. Here, CORDEX HR further exhibits the largest differences to the GCM ensembles and also often larger deviations from E-OBS. These latter regions are either characterized by complex and steep topography (e.g. the Alps and the Pyrenees), large fraction of coastal areas and/or by relatively dry environments dominated by precipitation of convective nature (particularly for the warmer months). These factors most likely play important roles for the larger differences seen between the low resolution CMIP GCMs and the higher resolution PRIMAVERA GCMs and CORDEX RCMs, as well as contributing to larger uncertainties in, and lower quality and representativeness of, observational data. In contrast, in almost all seasons over the British Isles, the CORDEX HR biases in total precipitation compared to E-OBS are among the smallest with respect to the other ensembles (the difference in the shape is similar). Finally, it is noted that for all regions PRIMAVERA HR and CORDEX LR give comparable distributions as they are of similar resolution.

To summarize, we can conclude that, in comparison to E-OBS, most model ensembles exhibit larger contributions for most precipitation intensities, but most consistent for low (< ca 3 mm day$^{-1}$) and moderate-to-high (> ca 10 mm day$^{-1}$). The larger contributions occur predominantly in DJF while in summer there are often lower contributions than in E-OBS for moderate intensities (leading to smaller biases in total means). In general, the CORDEX ensembles, and most often also PRIMAVERA, show a shift towards larger contributions from higher intensities compared to CMIP ensembles, especially in areas with complex orography as in the Alps. The higher model grid resolution does not always lead to improvements, i.e. closer agreements to E-OBS. However, it is worth re-emphasizing that the quality of E-OBS observations can be significantly lower in certain regions (e.g. mountainous areas or areas with

low density of precipitation gauges) and seasons (especially in wintertime when the fraction of snowfall
is largest which is more sensitive to wind induced undercatch) (Prein and Gobiet, 2017; Herrera et al.,
2019), thus complicating the assessment of model behaviour in comparison to observations. To further
highlight this issue, we have included an ASoP analysis for the Scandinavia region (Fig. S3) including a
regional high-quality high-resolution gridded observational data set; NGCD (Lussana et al., 2018). In
both DJF and JJA, the model ensembles still overestimate contributions from the bulk of the intensity
distribution; however, NGCD has higher contributions from low intensities compared to E-OBS,
reducing the model ensemble bias. More interestingly, NGCD shifts towards larger contributions for
high intensities, > 10 mm day$^{-1}$, in effect lending more credibility to the CORDEX HR ensemble and
less to the others.

### 3.1.3 Effect of grid resolutions – a one-to-one comparison

For multi-model ensembles, the sensitivity to model grid resolutions can generally only be assessed
qualitatively since other aspects, such as differences in model formulation, also contribute to differences
in model performance. In other words, it cannot be definitely stated to what extent differences in
performance comes from higher resolution or from other differences in the model code. For the
PRIMAVERA models, however, it is possible to directly compare low- and high-resolution model
versions. In CORDEX ensembles this is also possible to some extent for a few models where low- and
high-resolution versions of RCMs have been forced by the same parent GCMs. This is the case for nine
RCM-GCM combinations (6 different RCMs driven by 4 different GCMs). Note that, in contrast to
PRIMAVERA, CORDEX LR-HR "pairs" may not use the same version of the common model, which
could also influence the results in addition to change in grid resolution. Further, the magnitude of the
grid resolution change (the *delta* value) is the same for CORDEX models (*delta*=4), while for
PRIMAVERA models it varies between approximately 2 and 5. Figure 5 shows the one-to-one
comparison for DJF and JJA for selected regions. For CORDEX models the high-resolution model
versions generally generate, in both seasons, larger contributions from precipitation intensities above ca
10 mm day$^{-1}$. This is sometimes accompanied by lower contributions from lower rates as seen for
example in Scandinavia and the Alps in DJF. Similar results are seen for PRIMAVERA although not as

consistently; e.g. over the British Isles and the Alps in JJA about half the models show increased contributions in the HR models over the bulk part, the other half showing instead lower contributions (although for higher rates most HR models show larger contributions). In fact, for many regions there is a larger spread in JJA within each model ensemble and also between the individual LR versus HR responses compared to DJF. It could be argued that this effect is related to precipitation events being of more convective nature in summer and thus larger sensitivity to model grid resolution as well as model physics. In winter, CORDEX RCMs are to a larger extent being influenced by the forcing GCMs and therefore, as there is only four different GCMs used in the nine RCM-GCM combinations shown here, tends to exhibit more similar responses in this season.

## 3.2 Selected precipitation-based indices

### 3.2.1 Model ensemble comparison

Figure 6 shows the number of precipitation days (RR1, Table 3) as simulated by all models for each PRUDENCE region. The number of precipitation days does not differ much between the model ensembles. There are clear differences between individual models, but it is difficult to establish any significant differences between the model ensembles. This is the case both for regions with a higher occurrence of precipitation days (e.g. SC) and regions with fewer precipitation days (e.g. IP). All models show about the same number of precipitation events over the whole year, which may suggest that the large-scale weather patterns are not influenced that much by higher resolution; also, when looking at individual seasons the differences between ensembles are small (Fig. S4). Note, however, that the large-scale circulation in the RCMs to a large extent is governed by the driving GCM which have typical resolutions of around 200 km. Interpolating the data to a common grid prior to analysis does not have a large impact on RR1 (Fig. S5). Most models overestimate the number of precipitation days compared to observations. It is a well-known feature of climate models, particularly those with parameterized convection, that they tend to have too many wet days (e.g. Dai, 2006; Stephens et al., 2010).

The number of days with large precipitation amounts, above 10 mm day$^{-1}$ and 20 mm day$^{-1}$, become
more frequent with higher model resolution. For example, the number of days with precipitation over 20
mm (R20mm, Table 3) increases from just a few in CMIP5 to 5-10, or even more, in CORDEX HR
(Fig. 7). The 10$^{th}$ to 90$^{th}$ inter-percentile range increases, due to a larger increase in the 90$^{th}$ percentile.
Generally, the spread is larger for models with high resolution. This could partly be explained by higher
number of data points in the high-resolution models (i.e. larger number of grid points); a high-resolution
model is more likely to better represent the spatial variations of precipitation within a region while in
coarser scale models precipitation fields are smoother due to fewer grid points. The differences between
resolutions remain, however, also when all data are interpolated to two common grids of 0.5°×0.5° and
2°×2° resolutions; the median and spread also remain  similar in all ensembles. In small regions such as
AL the coarsest grid gives to few points, which means that it's difficult to calculate the 10th and 90th
percentiles. The spread in CORDEX HR increases when interpolated to 2°×2° because the points with
high values are not balanced by as many points close to the median (a 0.5°×0.5° grid contains 16 times
more points than a 2°×2° grid). Compared to E-OBS the average number of days with more than 20 mm
day$^{-1}$ is more accurately simulated in the high-resolution ensembles, but the spread is highly
exaggerated. The PRIMAVERA models have median values  similar to E-OBS and also a more similar
spread. The signal is the same for the individual seasons, but less pronounced since the potential
number of days is smaller when divided over four seasons instead of counted over the whole year (Fig
S6). The effect of resolution is therefore clearest in the season where most days occur, which means
winter in western Europe and summer in central Europe.

The fact that the number of wet days is similar between LR and HR models (Fig. 6) but with increased
frequency of (heavy) precipitation in HR models (Fig. 7) suggests that, for the latter, the precipitation
intensity on the wet days is higher. This is shown in the simple precipitation intensity index (SDII,
Table 3, Fig. 8). SDII is indeed affected by resolution, at least between CMIP5/6 and CORDEX; the wet
day average precipitation is larger in the HR simulations compared to LR models, and also the intra-
model spread (spread between models within the ensemble) is larger. For all regions, SDII is higher in
the HR models. Perhaps, the relative increase in SDII is higher in regions with large spatial variations
(for example because of complex orography or coastlines) such as IP and AL. The median SDII values
in high-resolution models are in all regions closer to E-OBS than the low-resolution models, even
though the model spread is generally larger in the climate models than in E-OBS. The differences
between ensembles remain both for the median and the spread when the data are regridded to common
grids. Also, for individual seasons it is clear that SDII increases with higher resolution, but the SDII
values do not vary much with season (Fig. S7).

The higher intensities for extreme precipitation in high-resolution models compared to low-resolution
models are also seen in the maximum one-day (Rx1day, Table 3, Fig. 9) and maximum five-day
precipitation (not shown).  There is a clear increase in both intensities and intra model spread in the
high-resolution models. It can be discussed if this increase is an improvement since the CORDEX HR
models give a maximum one-day precipitation that is significantly larger than E-OBS. On the other
hand, it can be discussed if E-OBS is able to reliably represent these extremes (Hofstra et al., 2009;
Prein and Gobiet, 2017). The medians and the spreads remain more or less the same also when
regridded to common grids. In small regions such as AL the spread is reduced because the number of
data points is small when regridded to a coarse grid. In regions with large spatial variations (e.g.
between coast and mountain) such as IP the spread increases because high values are not balanced by as
many points with values close to the median.  In winter the effect of higher resolution is mainly seen in
regions with complex topography, while in summer there is a clear signal in all regions (Fig 10). This
reflects that higher resolution makes the largest difference in complex topography and for convective
precipitation events.
**3.2.2 One-to-one comparison**
We let the mid-Europe region (ME) represent the whole domain, as the same conclusions can be made
for all regions, only with small differences in the number of models that give significant differences. A
one-to-one comparison is made of the selected indices for the models where there is both a low and a
high grid resolution version (Fig. 11). The LR and HR versions are compared with a Welsh's t-test
(Welsh, 1947) at the 0.05 significance level to see if the simulated indices are significantly different.
This corroborates the analysis above, and adds further detail by quantifying the differences.

Although the difference in the number of precipitation days (RR1, Fig. 11, top row) is significant for
most models it is not clear how it is affected by resolution. The differences are small, mainly within ±10
days year$^{-1}$, in some cases negative and in some positive. The differences between models are larger
than the differences between resolutions. It is clear, however, that all models overestimate the number
of precipitation days compared to E-OBS. This is true also when the data is regridded to common grids,
but three models and E-OBS get insignificant differences when regridded to 2°×2° instead of only one
model at the native grids.

The number of days with precipitation more than 20 mm (R20mm, Fig. 11, second row) is significantly
different between HR and LR for all models and E-OBS. For the CORDEX models R20mm is higher in
most HR versions, while the difference is less clear in the PRIMAVERA models. All simulations with
the RCA4 RCM, regardless of the driving GCM, clearly show higher R20mm in the HR version
compared to the LR versions, which indicates that the difference in the index mainly is a result of the
changed grid resolution in the RCM. The differences between LR and HR remain also when regridded
to common grids which means that this is an effect of differences in model physics. CORDEX LR is
close to E-OBS, while CORDEX HR generally overestimates R20mm.

The simple precipitation intensity index (SDII, Fig. 11, third row) is significantly different in one out of
four PRIMAVERA models and four out of nine CORDEX models. Differences are small, tenths of mm
day$^{-1}$, for most models. Most significant differences disappear when regridded to 0.5°×0.5° and all
disappear when regridded to 2°×2° suggesting that the resolution does not affect SDII much in these
model pairs. We still see a difference between CMIP GCMs and CORDEX RCMs (cf. Fig 8).

The maximum one-day precipitation (Rx1day, Fig. 11, bottom row) is significantly different in the HR
version in all but one model (a PRIMAVERA model). The HR versions have higher precipitation values
and larger spread in all but two PRIMAVERA models and one CORDEX model. Especially the
CORDEX HR models have a higher maximum one-day precipitation. This seems to be driven by the
RCM rather than the driving GCM. As an example, three RCMs are forced with the MPI-ESM-LR
GCM. When forced by this GCM the Rx1day in the CCLM4-8-17 RCM is lower in the HR version,
while in REMO2009 and RCA4 HR RCMs Rx1day is higher. In RCA4 the difference is particularly
large, regardless of the driving GCM. That the differences result from differences in model physics is
supported by the fact that the differences remain also when the data is regridded to common grids.
The one-to-one comparison of selected indices shows that there are significant differences between the
LR and HR models and that these are results of differences in model performance and not only the
number of data points. It also shows that for some indices the largest difference occurs between
CMIP5/6 and PRIMAVERA HR, rather than between PRIMAVERA and CORDEX. This means that
some of the differences seen in Figures 6-10 are not as clear in figure 11. The comparison also shows
that even though there are significant differences between LR and HR it is for some cases difficult to
establish significant differences between two ensembles since the difference between two models are
often larger than between the LR and HR version of the same model.
It should be noted that the CORDEX RCMs are not always run with the same model version in the LR
and HR simulations. Model differences could thus explain some of the differences between LR and HR.
Since we don't have LR and HR simulations with all model versions we can't quantify this effect, only
acknowledge it. It should also be noted that the difference in horizontal grid spacing varies between
models. For CORDEX RCMs the resolution *delta* (LR/HR) is always 4 (50 km/12.5 km), but for
PRIMAVERA it varies between 2 and 5. The *delta* value is larger in CORDEX than in most
PRIMAVERA models, which could potentially mean that the effect of resolution is overestimated for
the CORDEX RCMs. Figure 12 shows how the absolute differences in RR1, R20mm, SDII and Rx1day
between the LR and HR version of the PRIMAVERA and CORDEX models described above correlates
to the *delta* value in the ME region. There is no clear relation between the *delta* value and the size of the
difference. CORDEX models that all have the same *delta* value span from small to large differences.
The spread between PRIMAVERA models is also quite large. This again suggests that the response of a
model to increased resolution depends on the model itself and not only on the magnitude of the
resolution change.

## 4 Discussion and conclusions

This study investigates the importance of model resolution on the simulated precipitation in Europe.
The aim is to investigate the differences between models and model ensembles, but also to evaluate
their performance compared to gridded observations. In a similar study Demory et al. (2020) compare
PRIMAVERA models with CORDEX LR and CORDEX HR. They conclude that CORDEX
indisputably improves the data from the driving CMIP5 models, but that the differences between
CORDEX LR and PRIMAVERA are generally small. Both ensembles perform well, but tend to
overestimate precipitation in winter and spring. The largest differences between the ensembles are for
high precipitation intensities, in especially summer, where PRIMAVERA gives less heavy precipitation
which makes it agree more with observations than CORDEX. Iles et al. (2020) compare the effect of
resolution on extreme precipitation in Europe in CMIP5 GCMs and CORDEX RCMs. They conclude
that high resolution models systematically produce higher frequencies of high-intensity precipitation
events. Our interpretation of this, given the results in our study, is that in some cases also the
overestimation of precipitation compared to E-OBS increases with higher resolution. The findings in
this study support the conclusions from the above-mentioned studies, and add details based on a wider
range of model ensembles and precipitation metrics. The fact that we come to the same conclusions as
Iles et al. (2019) and Demory et al (2020) with slightly different methods give strength to these
conclusions.
The ASoP analysis in this study shows that all model ensembles have larger contributions from heavy
precipitation in winter compared to E-OBS, and that the higher values become most prominent for the
ensemble with the highest grid resolution, CORDEX HR. The biases compared to E-OBS are generally
smaller in summer. The PRIMAVERA ensemble is in good agreement with observations and has
smaller bias than CORDEX for many regions. CMIP5 and CMIP6 mostly underestimate contributions
from moderate-to-high precipitation intensities in summer while overestimating low-intensity events.
Overall, in the summer season, the spread is large between ensembles and between models within the
ensembles. This is indicative of large uncertainties which are most likely related to uncertainties in how
models are able to treat smaller scale precipitation events involving convection. With respect to E-OBS,
the ASoP results partly show that higher horizontal grid resolution does not necessarily mean better.
However, in coastal regions and regions with steep or complex topography there are uncertainties in
both models and observations. Particularly in winter observations suffer from undercatch when
precipitation falls as snow during windy conditions and in summer, smaller scale convective
precipitation may be smoothed considerably or missed completely by ground rain gauges (which E-
OBS is based on). E-OBS is not based on the full network of rain gauges in all countries, which could
also lead to undercatch. Therefore, it is not always obvious which model or ensemble of models is
closest to reality. When compared to NGDC, a regional data set of high-quality, the difference between
CORDEX HR and observations is reduced, which gives more confidence to the high-resolution model
results.

It is clear that the horizontal resolution of a model has a large effect on precipitation, mostly on the
heavier precipitation and in areas with complex and steep orography. The number of precipitation days
does not depend much on resolution as this is mostly depending on large scale weather patterns and not
so much on local topography and convection. For heavy precipitation events, which often are more local
and short-lived in character, model resolution is more important. The high-resolution models better
resolve such events and distinguish better between different parts of a region. Thus, extreme
precipitation is more intense and more frequent in the HR models compared to the LR models in this
study. With the same amount of wet days this means that precipitation intensifies so that the wet days
get wetter. The largest impact of increased model scale resolution on precipitation is most evident for
the coarser scale models; increasing the resolution from CMIP5/6 to PRIMAVERA HR has a greater
effect than increasing from CORDEX LR/PRIMAVERA HR to CORDEX HR. This does not, however,
mean that increased resolution gets less and less worthwhile; further refining the grid until convection-
permitting resolutions are reached (less than ~5 km grid spacing), in which case convection
parameterizations may be turned off, has a large positive effect (e.g. Prein et al. 2015). This is not
shown here as the smallest grid spacing in models in this study is 12.5 km. The effect of higher
resolution is seen in regions with small amounts of precipitation as well as regions with high amounts of
precipitation, and in regions with small and large geographical differences. The higher percentiles
change more than the low percentiles for all studied indices. Increasing resolution has about the same
effect on both GCMs and RCMs, furthermore GCMs and RCMs of comparable resolution simulate
comparable precipitation climates, even though PRIMAVERA is often drier than CORDEX.

It is worth to note that the differences between RCM simulations, and how they respond to differences
in resolution, may very well be explained by the driving GCM and the state of the atmospheric general
circulation in them (Kjellström et al., 2018; Sørland et al., 2018; Vautard et al., 2020). Higher resolution
is expected to give a better described and more detailed climate, with for example deeper cyclones and
more intense local showers; in a sense with more pronounced weather events. If two models are in
different states, for example when it comes to where storm tracks cross Europe, and if these states are
pronounced, that may lead to even larger model differences. Instead of a weak storm track in the south
and a weak storm track in the north in the low-resolution model, we may now instead have strong storm
tracks, which mean that the difference between the models increases. Still, the largest differences are
seen in the CORDEX ensemble where the LR and HR models are run with the same coarse resolution
GCM. This suggests that (regional) model resolution and performance is what determines high
precipitation rates, rather than the driving GCM. To fully answer that would require an analysis of the
circulation patterns in the different models. This is not done here, but should be a topic for further
studies.

The differences between LR and HR largely remain also when the results are regridded to common
grids of 0.5°×0.5° and 2°×2° which means that the HR version performs differently than the LR version
of the same model, mainly because of better representations of topography and convection. The largest
seasonal differences are seen for the heavy precipitation (R20mm, Rx1day). Heavy precipitation events

usually occur locally in summer which makes it more sensitive to model resolution. Difference in resolution has a larger impact on heavy precipitation in summer than in winter.

Higher resolution does not necessarily mean better results. If a model is already too wet the increase in heavy precipitation that is induced by the higher resolution means that the HR version agrees less with observations than the LR version. For the individual model it is possible to quantify the difference and improvement between LR and HR. On the ensemble level this is more difficult. The difference between different models is often larger than between LR and HR versions of the same model. In this sense the quality of an ensemble is depending more on the models it consists of rather than the average resolution of the ensemble. Furthermore, when downscaling with an RCM, the simulated extreme precipitation, and the differences between GCM and RCM, depends more on the used RCM and less on the down-scaling itself, especially for heavy precipitation and particularly in summer.

## Acknowledgements

The authors would like to thank Ségolène Berthou and two anonymous reviewers for giving valuable comments on the manuscript. This work has been funded by the PRIMAVERA project, which is funded by the European Union's Horizon 2020 programme, Grant Agreement no. 641727PRIMAVERA. This work used JASMIN, the UK collaborative data analysis facility. Some analyses were performed on the Swedish climate computing resource Bi provided by the Swedish National Infrastructure for Computing (SNIC) at the Swedish National Supercomputing Centre (NSC) at Linköping University. We acknowledge the E-OBS dataset from the EU-FP6 project UERRA (http://www.uerra.eu) and the Copernicus Climate Change Service, and the data providers in the ECA&D project (https://www.ecad.eu). We thank the modelling groups that run models and provide data within CMIP5, CMIP6, PRIMAVERA and CORDEX.

Data: The data are stored on the Jasmin infrastructure, http://www.ceda.ac.uk/projects/jasmin/. The simulations are part of the High Resolution Model Intercomparison project (HiResMIP) and will be

uploaded to the ESGF: https://esgf-node.llnl.gov. Scripts for analysing the data will be available from
the corresponding authors upon reasonable request.

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

## Tables

| Ensemble | Model | Contact institute | Atmospheric grid spacing |
|----------|-------|-------------------|--------------------------|
| CMIP5 | ACCESS1-0 | Commonwealth Scientific and Industrial Research Organisation, Australia, and Bureau of Meteorology | N96 |
| CMIP5 | ACESS1-3 | Commonwealth Scientific and Industrial Research Organisation, Australia, and Bureau of Meteorology | N96 |
| CMIP5 | CanESM2 | Canadian Centre for Climate Modelling and Analysis | T63 |
| CMIP5 | CMCC-CESM | Centro Euro-Mediterraneo per i Cambiamenti Climatici | 96x48 |
| CMIP5 | CMCC-CM | Centro Euro-Mediterraneo per i Cambiamenti Climatici | 480x240 |
| CMIP5 | CMCC-CMS | Centro Euro-Mediterraneo per i Cambiamenti Climatici | 192x96 |
| CMIP5 | CSIRO-Mk3-6-0 | Australian Commonwealth Scientific and Industrial Research Organization (CSIRO) Marine and Atmospheric Research in collaboration with the Queensland Climate Change Centre of Excellence (QCCCE) | T63 |
| CMIP5 | FGOALS-g2 | Institute of Atmospheric Physics, Chinese Academy of Sciences and Tsinghua University | 128x60 |
| CMIP5 | GFDL-CM3 | NOAA Geophysical Fluid Dynamics Laboratory | 144x90 |
| CMIP5 | GFDL-ESM2G | NOAA Geophysical Fluid Dynamics Laboratory | 144x90 |
| CMIP5 | HadCM3 | Met Office Hadley Centre | 96x73 |
| CMIP5 | HadGEM2-CC | Met Office Hadley Centre | N96 |
| CMIP5 | HadGEM2-ES | Met Office Hadley Centre | N96 |
| CMIP5 | IPSL-CM5A-LR | Institut Pierre Simon Laplace | 96x96 |
| CMIP5 | IPSL-CM5A-MR | Institut Pierre Simon Laplace | 144x143 |

| | | | |
|---|---|---|---|
| CMIP5 | MPI-ESM-LR | Max Planck Institute for Meteorology | T63 |
| CMIP5 | MPI-ESM-MR | Max Planck Institute for Meteorology | T63 |
| CMIP5 | NorESM1-M | Norwegian Climate Centre | 144x96 |
| CMIP6 | ACCESS-CM2 | Commonwealth Scientific and Industrial Research Organisation, Australia, and Bureau of Meteorology | 192x145 |
| CMIP6 | ACCESS-ESM1-5 | Commonwealth Scientific and Industrial Research Organisation, Australia, and Bureau of Meteorology | 192x145 |
| CMIP6 | CESM2-FV2 | The National Center for Atmospheric Research | 144x96 |
| CMIP6 | CESM2 | The National Center for Atmospheric Research | 288x192 |
| CMIP6 | CESM2-WACCM-FV2 | The National Center for Atmospheric Research | 144x96 |
| CMIP6 | CESM2-WACCM | The National Center for Atmospheric Research | 288x192 |
| CMIP6 | EC-Earth3 | EC-Earth-Consortium | 512x256 |
| CMIP6 | EC-Earth3-Veg | EC-Earth-Consortium | 512x256 |
| CMIP6 | GFDL-CM4 | NOAA Geophysical Fluid Dynamics Laboratory | 360x180 |
| CMIP6 | INM-CM4-8 | Institute for Numerical Mathematics, Russian Academy of Science | 180x120 |
| CMIP6 | INM-CM5-0 | Institute for Numerical Mathematics, Russian Academy of Science | 180x120 |
| CMIP6 | MIROC6 | Japan Agency for Marine-Earth Science and Technology, Atmosphere and Ocean Research Institute, The University of Tokyo, National Institute for Environmental Studies, RIKEN Center for Computational Science | T85 |
| CMIP6 | MPI-ESM-1-2-HAM | Max Planck Institute for Meteorology | 192x96 |
| CMIP6 | MPI-ESM1-2-LR | Max Planck Institute for Meteorology | 192x96 |
| CMIP6 | MRI-ESM2-0 | Meteorological Research Institute, Tsukuba | 320x160 |
| CMIP6 | NorCPM1 | Norwegian Climate Centre | 320x384 |
| CMIP6 | NorESM2-LM | Norwegian Climate Centre | 144x96 |
| CMIP6 | NorESM2-MM | Norwegian Climate Centre | 288x192 |
| CMIP6 | SAM0-UNICON | Seoul National University | 288x192 |
| PRIMAVERA | CNMR-CM6-1 | CNRM-CERFACS | 256x128 |
| PRIMAVERA | CNRM-CM6-1-HR | CNRM-CERFACS | 720x360 |
| PRIMAVERA | EC-Earth3 | EC-Earth-Consortium | 512x256 |

| PRIMAVERA | EC-Earth3-HR | EC-Earth-Consortium | 1024x512 |
|---|---|---|---|
| PRIMAVERA | IFS-HR | European Centre for Medium-Range Weather Forecasts | 720x360 |
| PRIMAVERA | IFS-LR | European Centre for Medium-Range Weather Forecasts | 360x180 |
| PRIMAVERA | HadGEM3-GC31-HM | Met Office Hadley Centre | 1024x720 |
| PRIMAVERA | HadGEM3-GC31-LM | Met Office Hadley Centre | 192x144 |
| PRIMAVERA | HadGEM3-GC31-MM | Met Office Hadley Centre | 432x324 |
| PRIMAVERA | MPIESM-1-2-HR | Max Planck Institute for Meteorology | 384x192 |
| PRIMAVERA | MPIESM-1-2-XR | Max Planck Institute for Meteorology | 768x384 |

**Table 1.** The GCM ensembles used in this study and the GCMs they consist of. Grid spacing is given in the same format as
in the meta data for each model.

| Institute | RCM | Driving GCM | | | | | | | | | |
|---|---|---|---|---|---|---|---|---|---|---|---|
| | | 1 | 2 | 3 | 4 | 5 | 6 | 7 | 8 | 9 | 10 |
| CLMcom | CCLM4-8-17 | x | x | | x | | x | | x | xo | |
| CNRM | ALADIN53 | | x | | | | | | | | |
| CNRM | ALADIN63 | | x | | | | | | | | |
| DMI | HIRHAM5 | | | | xo | | x | | | | x |
| GERICS | REMO2015 | x | x | | x | | x | | x | | x |
| IPSL | WRF331F | | | | | | | xo | | | |
| KNMI | RACMO22E | | | | xo | | o | | | | x |
| MPI-CSC | REMO2009 | | | | | | | | | xo | |
| SMHI | RCA4 | o | o | o | xo | o | xo | xo | o | xo | o |
| UHOH | WRF361H | | | | | | x | | | x | |
| HMS | ALADIN52 | | o | | | | | | | | |

**Table 2.** RCM GCM combinations used in this study. EURO-CORDEX simulations at 0.11° (~12.5 km) are marked with
"x" and at 0.44° (~50 km) are marked with "o". The driving GCMs are: 1) CanESM2, 2) CNRM-CM5, 3) CSIRO-Mk3-6-0,
4) EC-Earth, 5) GFDL-ESM2M, 6) HadGEM2-ES, 7) IPSL-CM5A-MR, 8) MIROC5, 9) MPI-ESM-LR, 10) NorESM1-M


| Short | Long name | Definition | Unit |
|---|---|---|---|

| name | | | |
|------|------|------|------|
| RR1 | Wet days index | Number of days with precipitation sum equal to or more than 1 mm | Days year$^{-1}$ |
| R20mm | Very heavy precipitation days index | Number of days with precipitation sum more than 20 mm | Days year$^{-1}$ |
| SDII | Simple daily intensity index | Average precipitation sum on days with precipitation sum equal to or above 1 mm | mm day$^{-1}$ |
| Rx1day | Highest one day precipitation amount | Precipitation amount on the day with highest amount | mm day$^{-1}$ |

**Table 3.** Definitions of indices


**Figures**

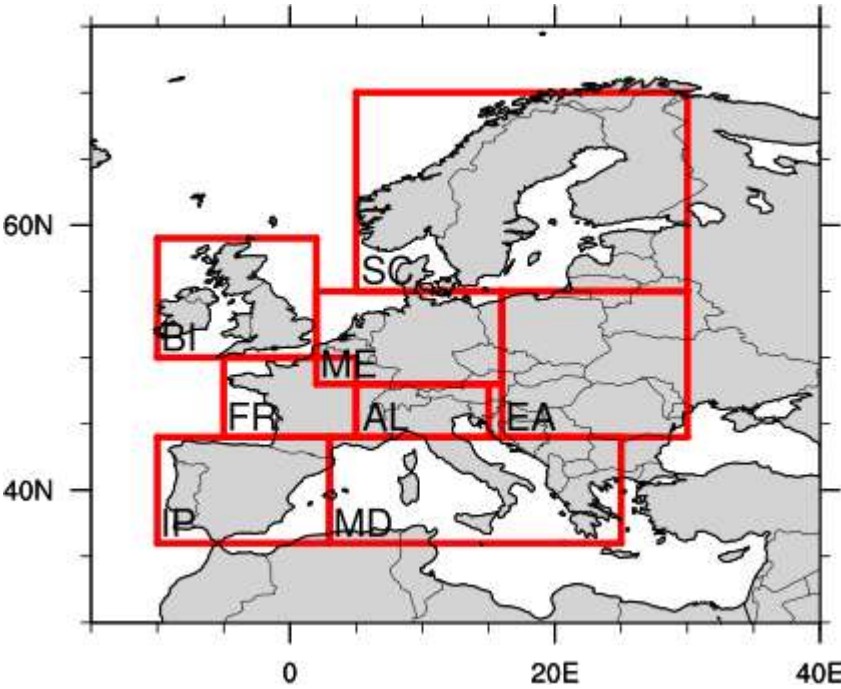


**Figure 1: The regions for which precipitation data is analysed: Scandinavia (SC), British Isles (BI), Mid-Europe (ME), France**
**(FR), The Alps (AL), Eastern Europe (EA), Iberian Peninsula (IP) and the Mediterranean (MD).**

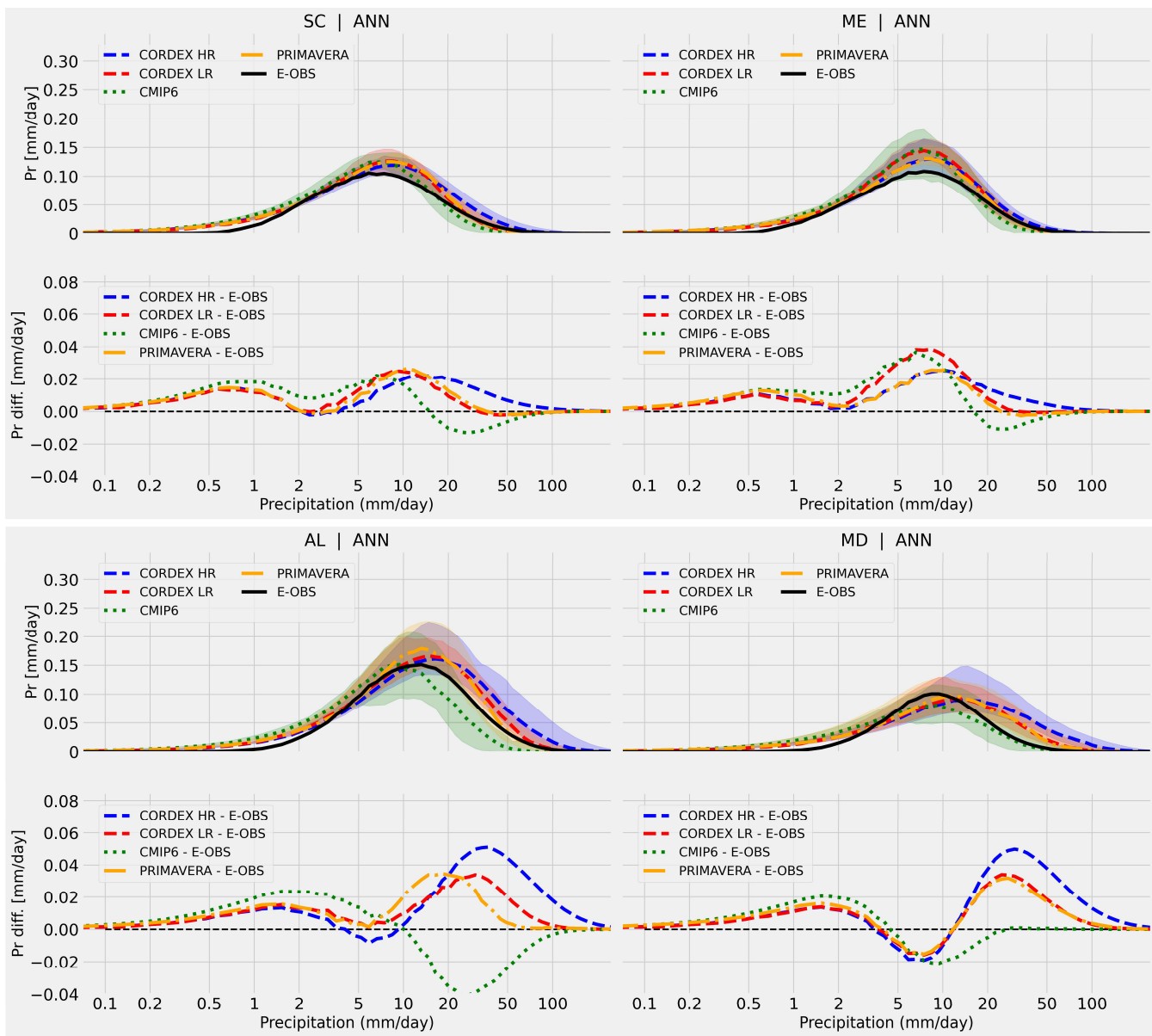

**Figure 2: The panels show the actual contribution (to the total median precipitation, y-axis) per precipitation intensity bin (x-axis),**
**based on annual (ANN) daily precipitation values in the CMIP6 (green dotted lines and shading), PRIMAVERA (orange dashed-**
**dotted lines and shading), CORDEX low resolution (red dashed lines and shading) and CORDEX high resolution (blue dashed**
**lines and shading) ensembles. The displayed regions are Scandinavia (SC, top left), mid-Europe (ME, top right), the Alps (AL,**
**bottom left) and the Mediterranean (MD, bottom right). Coloured shadings represent the 5-95 percentile range in respective**
**ensemble. Black solid lines are E-OBS (0.1º resolution) observations.**

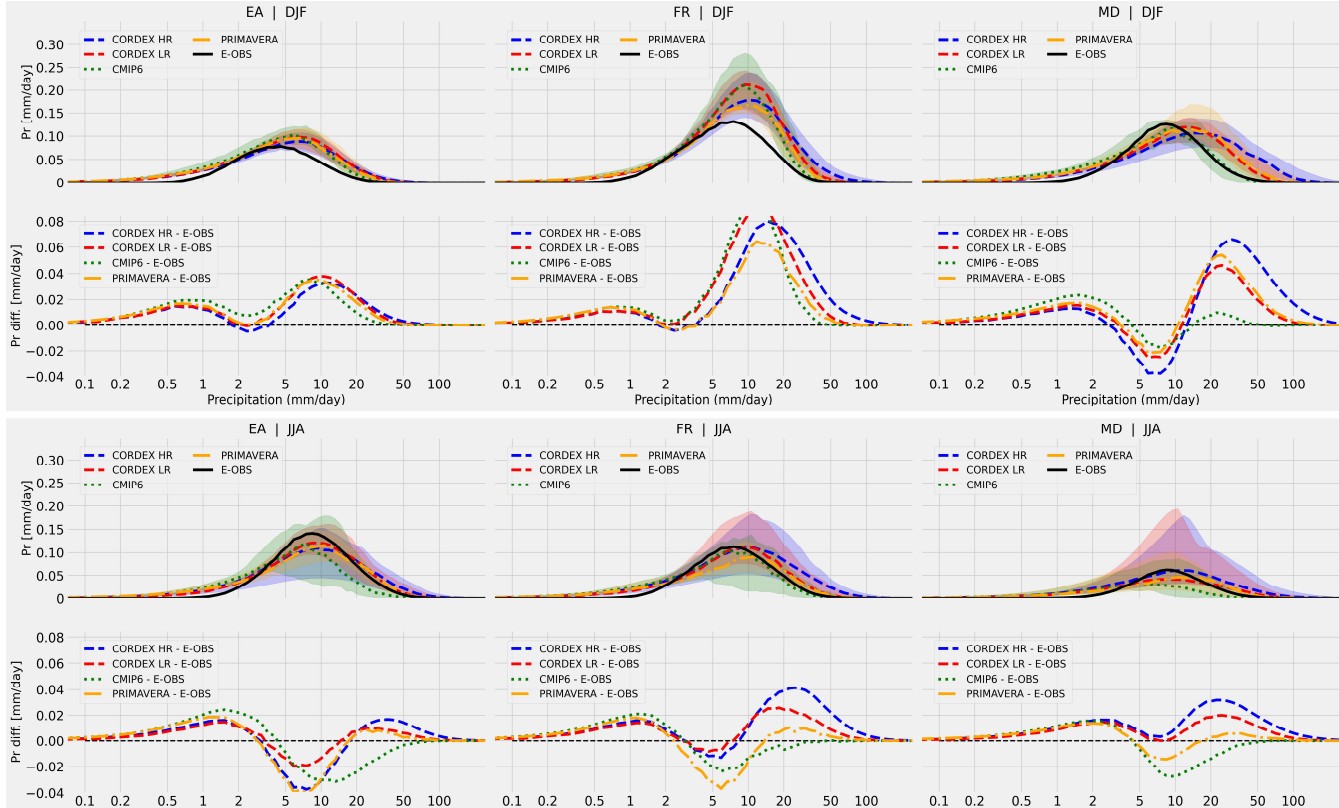


**Figure 3: Same as in Fig. 2 but for DJF (top row) and JJA (bottom row) daily precipitation values and for the eastern Europe (EA,**
**left), France (FR, middle) and the Mediterranean (MD, right) regions. Coloured shadings represent the 5-95 percentile range in**
**respective ensemble. Black solid lines are E-OBS (0.1º resolution) observations.**

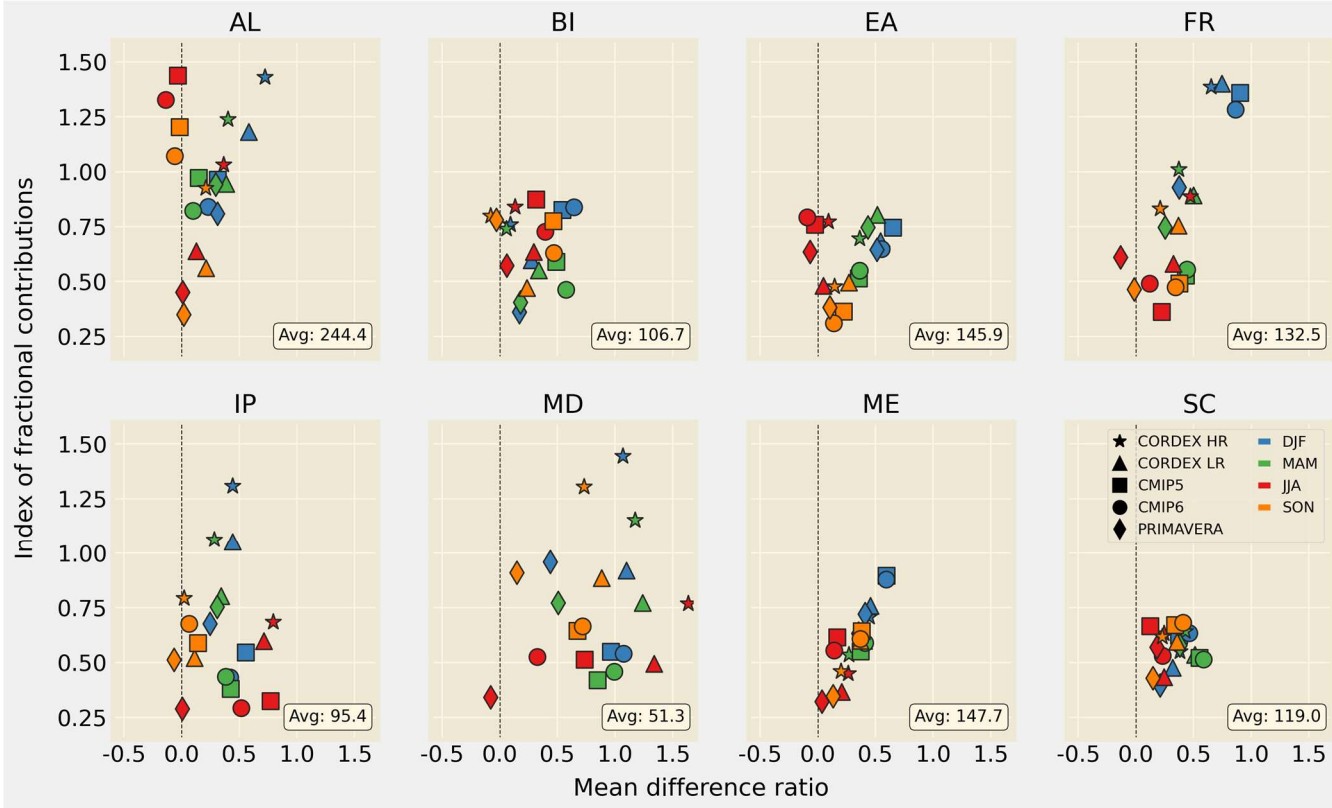


Figure 4: The index of fractional contributions (y-axis) plotted as a function of the fractional difference in seasonal total
precipitation (x-axis). E-OBS (0.1º resolution) is the reference data set and E-OBS average annual total precipitation (in mm year⁻
¹) is shown in lower right in each panel.

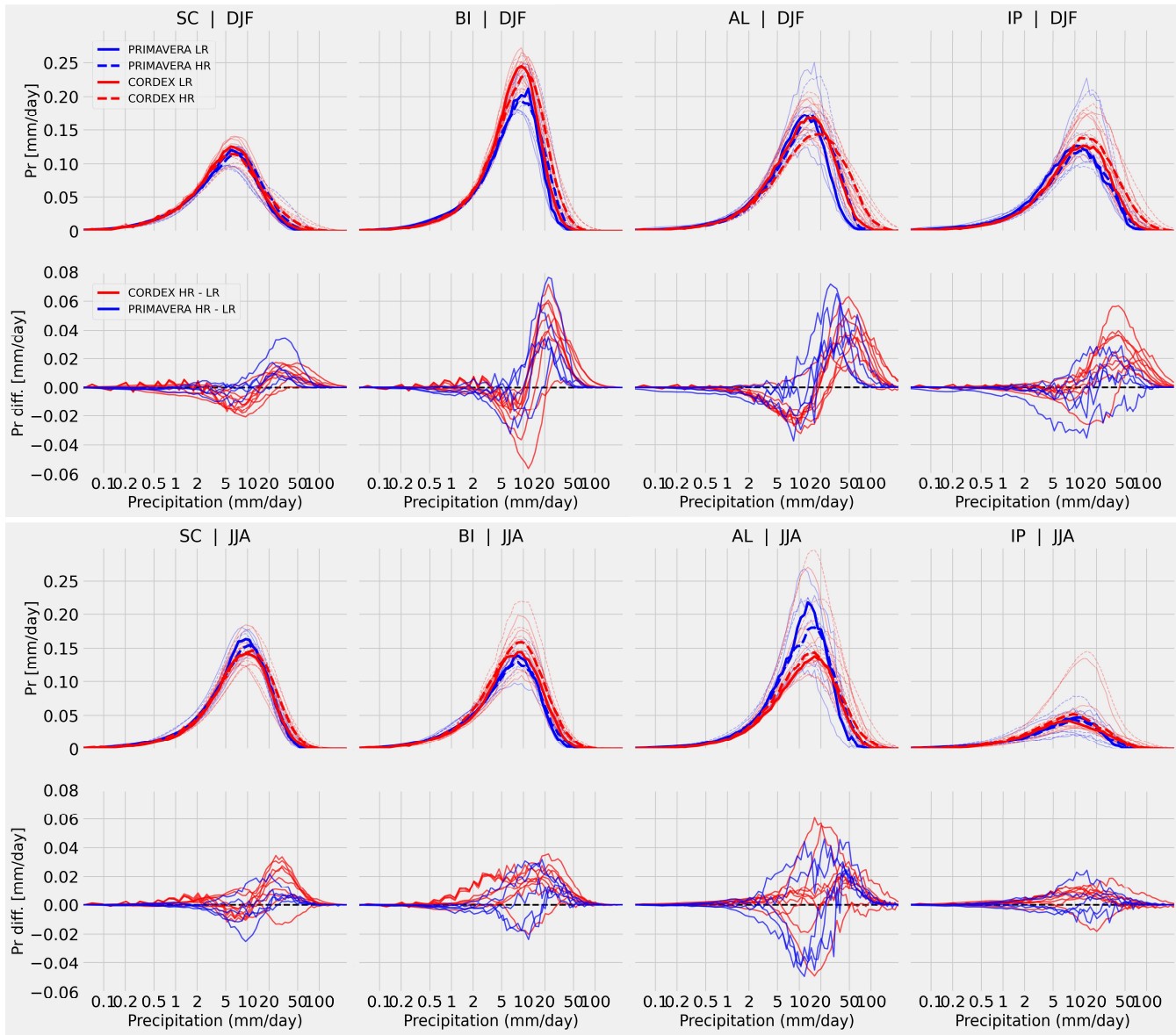

**Figure 5: The panels show the actual contribution (to the total mean precipitation, y-axis) per precipitation intensity bin (x-axis), based on DJF (top row) and JJA (bottom row) daily mean precipitation values in CORDEX and PRIMAVERA models for the Scandinavia (SC), British Isles (BI), the Alps (AL) and Iberian Peninsula (IP) regions. Thin lines in upper part of each panel represent each individual model while the thick lines represent the ensemble means. In the lower part of each panel each line represents differences between respective high- and low-resolution model pair.**

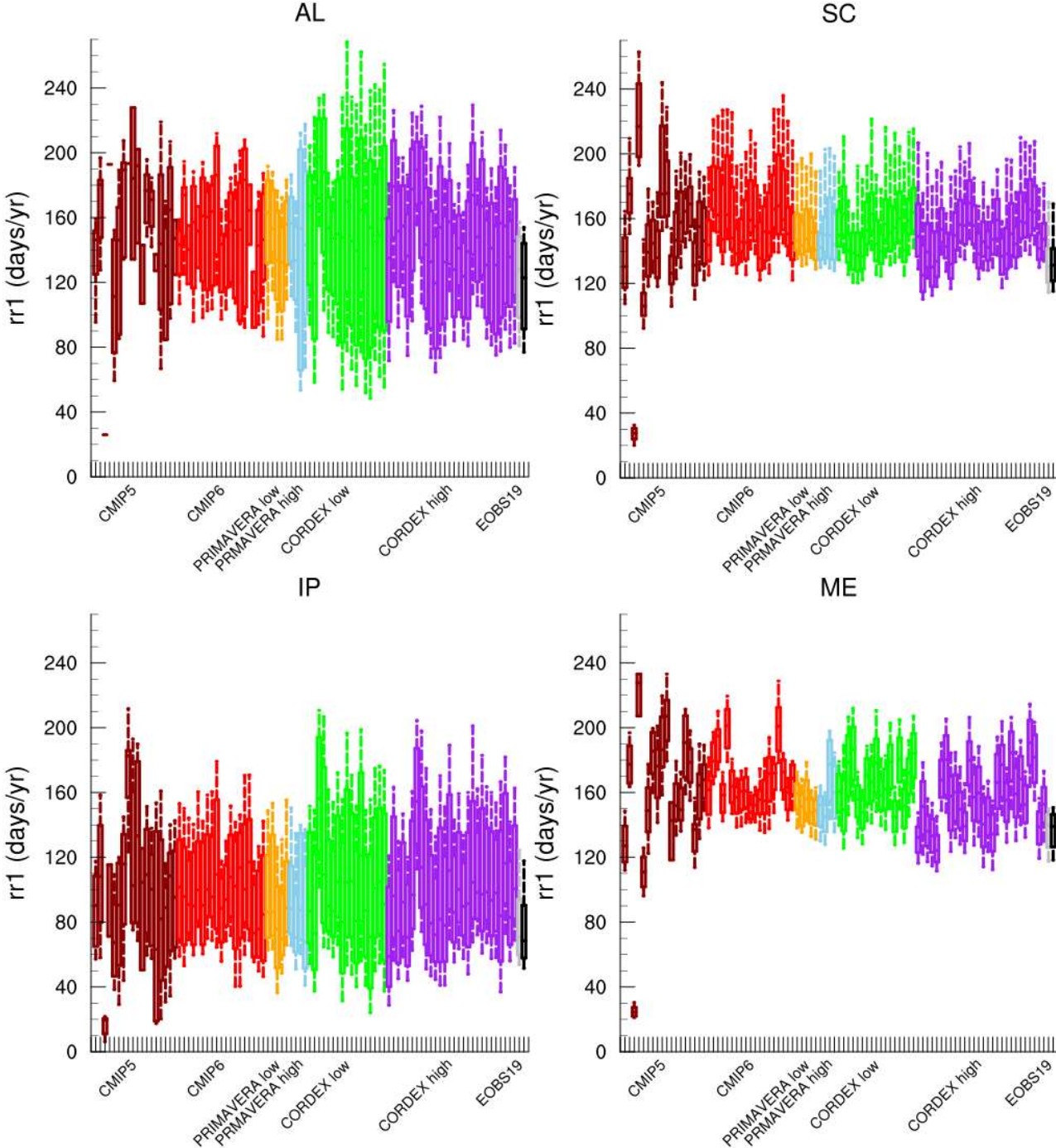

**Figure 6. Number of precipitation days (RR1 (days year⁻¹]) in the Alps (AL, top left), Scandinavia (SC, top right), the Iberian**
**Peninsula (IP, bottom left) and mid-Europe (ME, bottom right) for individual models in the CMIP5 (brown), CMIP6 (red),**
**PRIMAVERA LR (orange), PRIMAVERA HR (light blue), CORDEX LR (green) and CORDEX HR (purple) ensembles as well as**
**E-OBS at 28 (grey) and 11 km (black). Boxes mark the 25th and 75th percentile, with the median inside; whiskers go from the 10th**
**to the 90th percentile.**

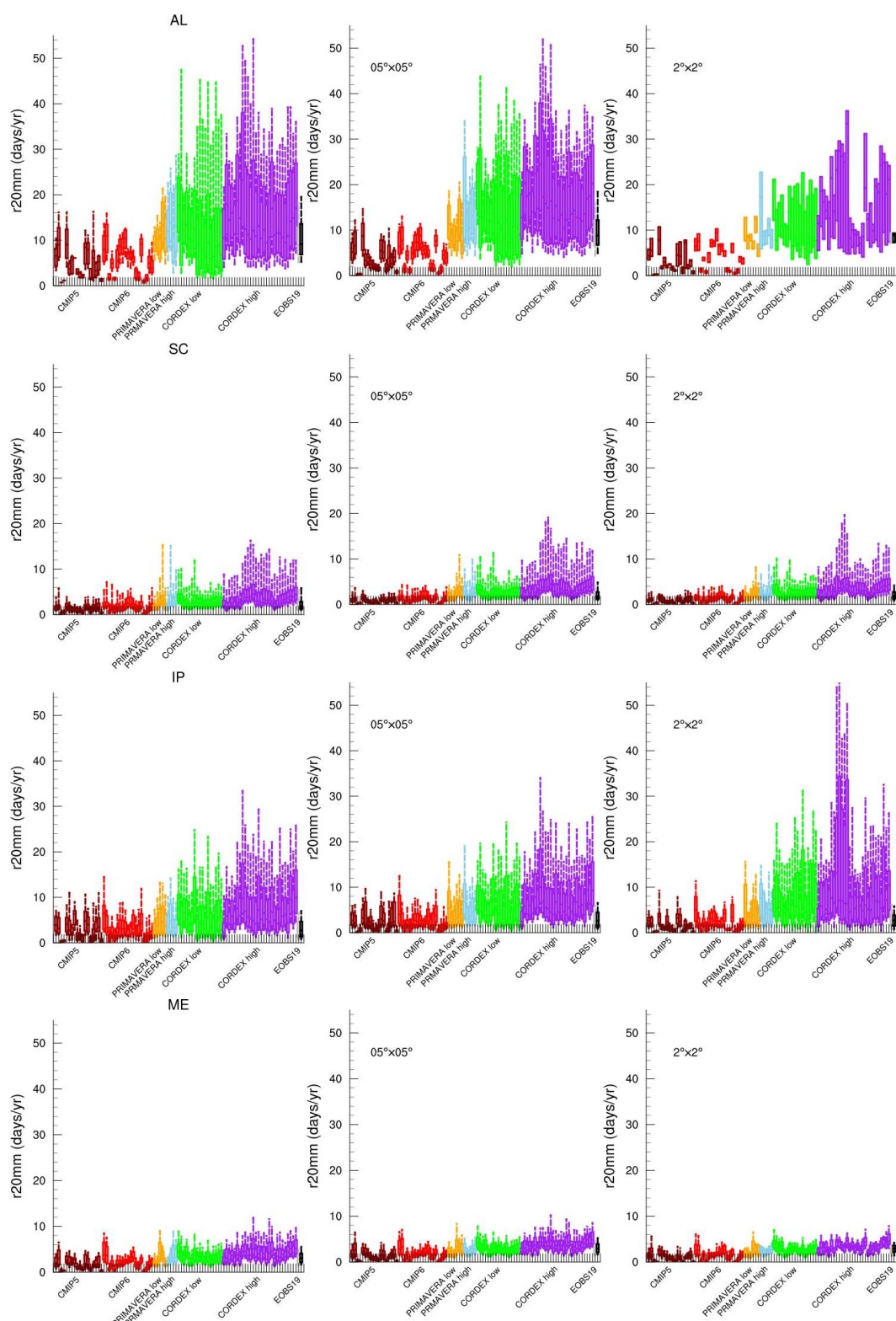


**Figure 7. Same as Figure 6 but for the number of days with precipitation amount over 20 mm (R20mm (days year$^{-1}$)). Left column:**
**model data on their original grids, centre column: all data regridded to 0.5°×0.5° grid, right column: all data regridded to  2°×2°**
**grid.**

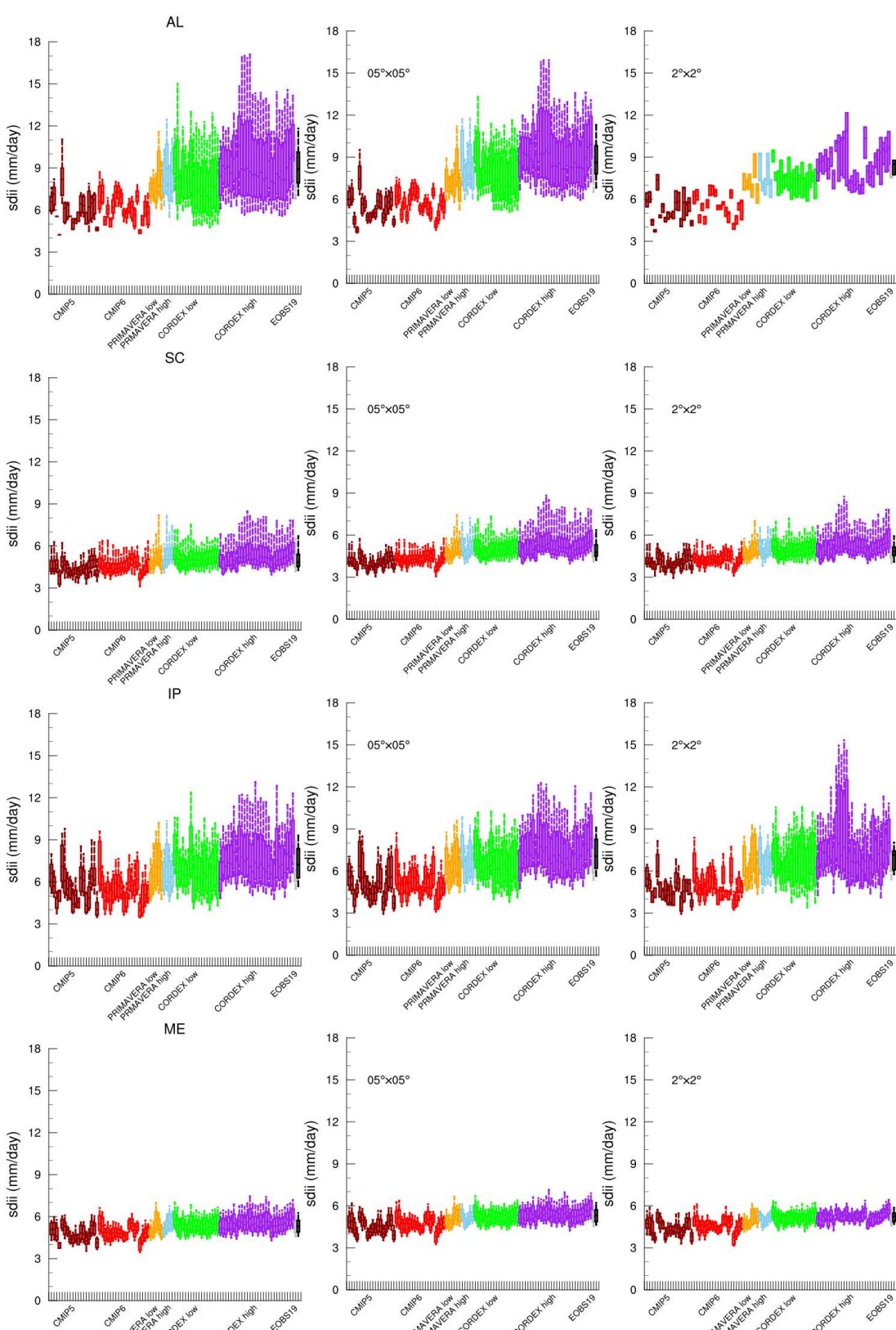


**Figure 8. Same as Figure 7 but for the simple precipitation intensity index (SDII (mm day$^{-1}$)).**

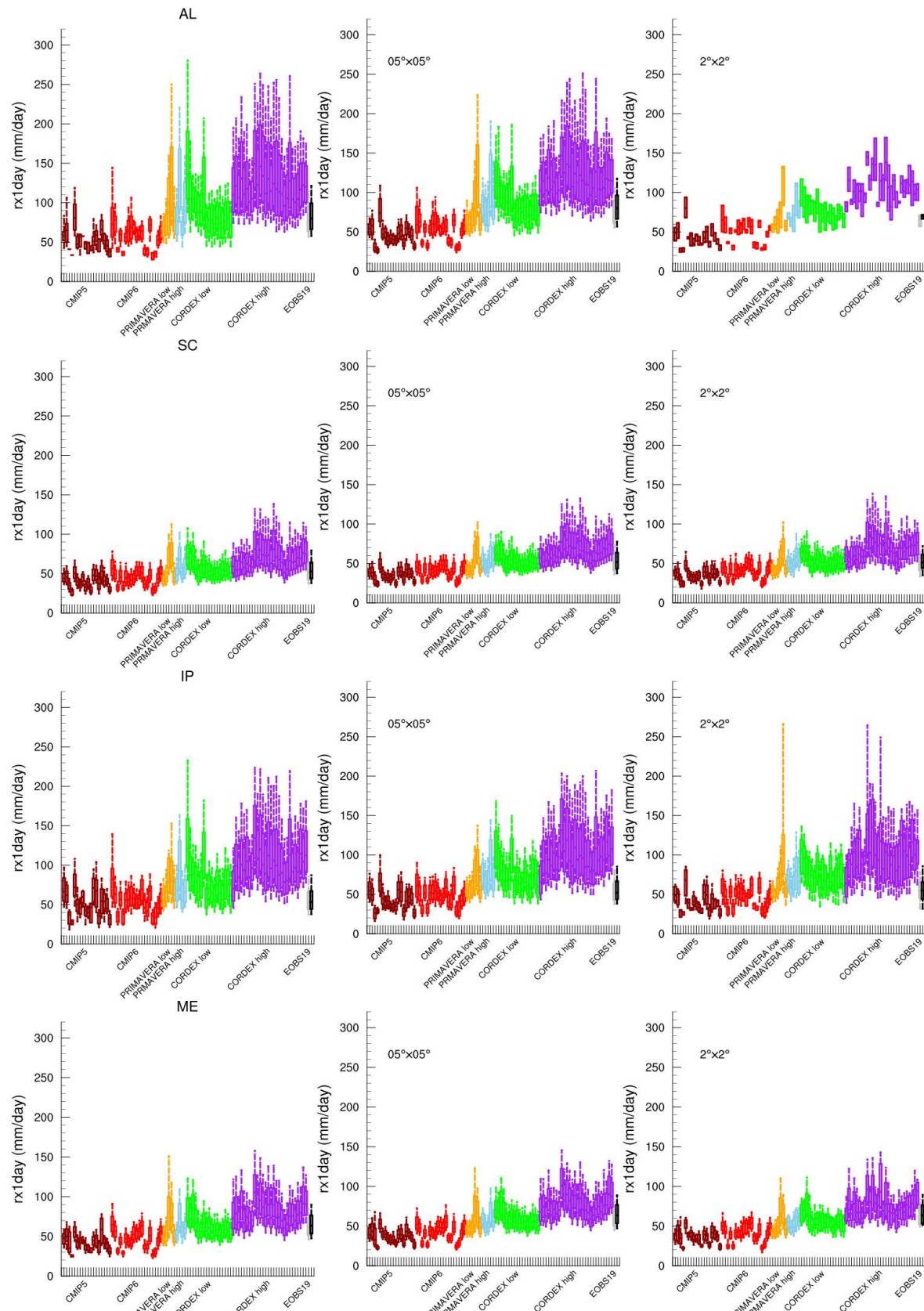


**Figure 9. Same as Figure 7 but for the maximum one day precipitation (Rx1day (mm day$^{-1}$)).**


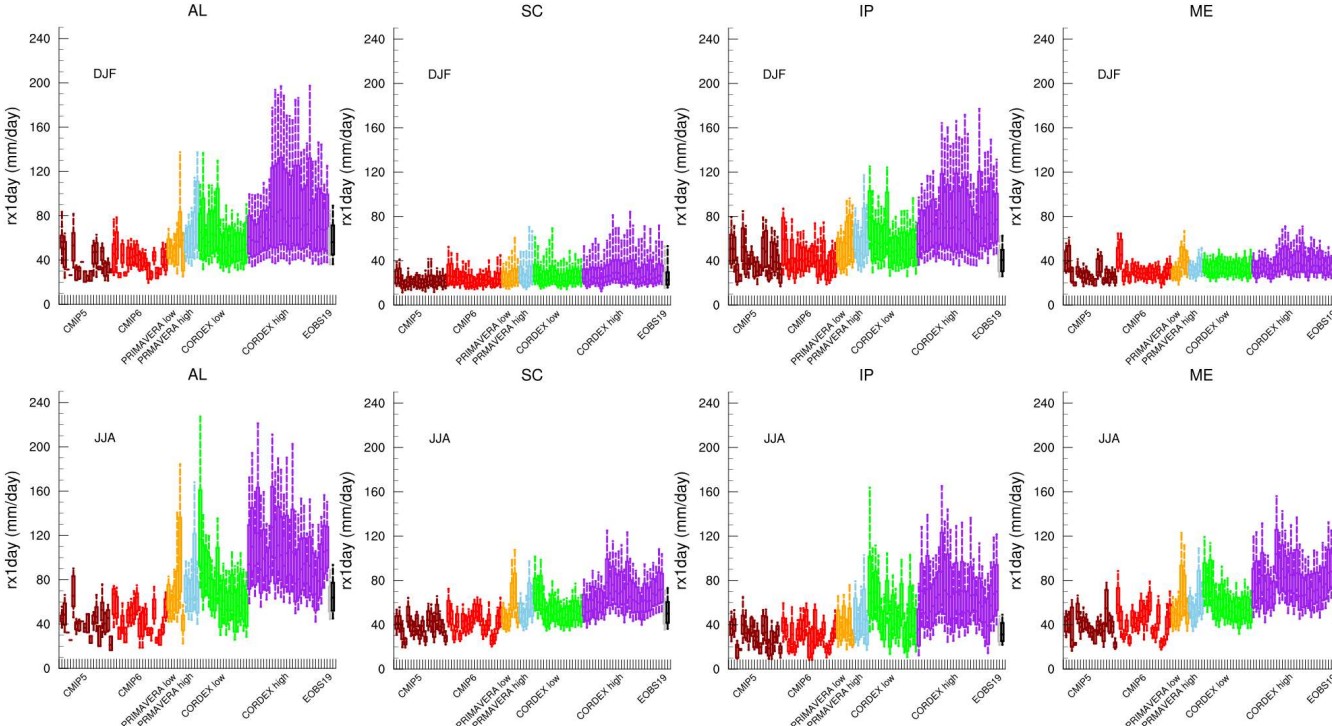


**Figure 10. Same as Figure 6 but for the maximum one-day precipitation (Rx1day (mm day⁻¹)), top row: winter (DJF), bottom row:**
**summer (JJA).**

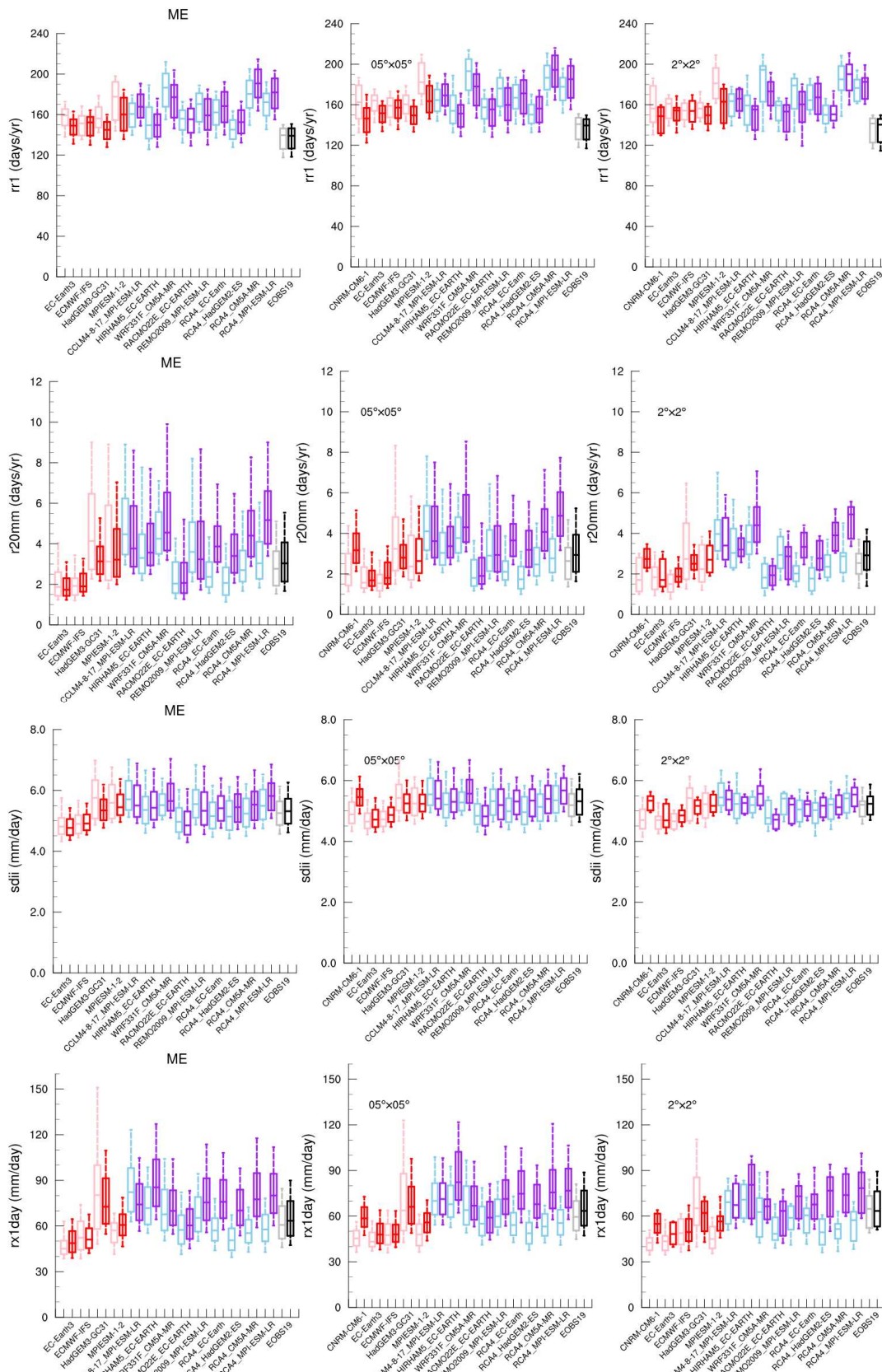


**Figure 11. Number of precipitation days (RR1 (days year$^{-1}$), first row), number of days with precipitation amount over 20 mm**
**(R20mm (days year$^{-1}$), second row), simple precipitation intensity index (SDII (mm day$^{-1}$), third row), maximum one day**
**precipitation (Rx1day (mm day$^{-1}$), fourth row) in the Mid-European region (ME) in the PRIMAVERA LR (pink) and HR (red)**
**models, CORDEX LR (light blue) and HR (purple) models as well as E-OBS LR (grey) and HR (black). Left column: model data**
**on their original grids, centre column: all data regridded to 0.5°×0.5° grid, right column: all data regridded to 2°×2° grid. Boxes**
**mark the 25$^{th}$ and 75$^{th}$ percentile, with the median inside; whiskers go from the 10$^{th}$ to the 90$^{th}$ percentile. If the the high-resolution**
**version of a model is significantly different from the low-resolution version this is marked with a vertical line in the high-resolution**
**boxes.**


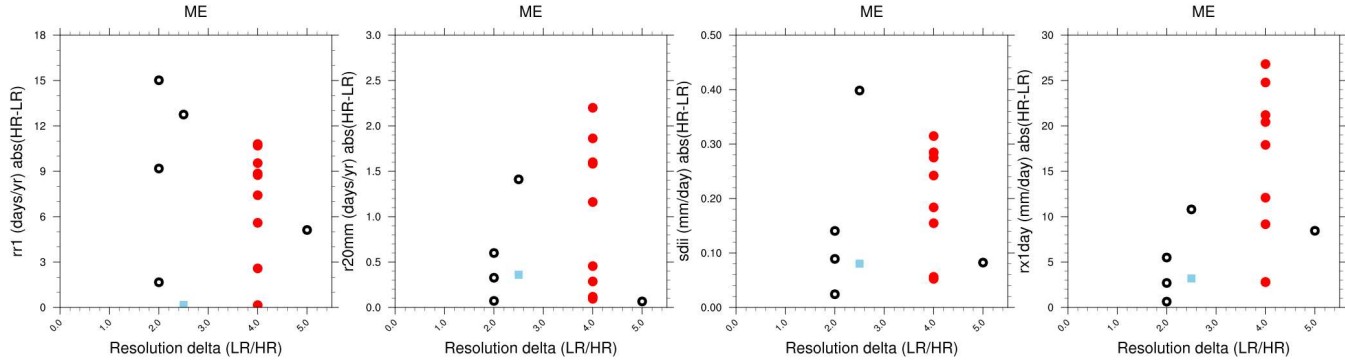


**Figure 12. Absolute difference between HR and LR version of PRIMAVERA (black rings), CORDEX (red circles) and E-OBS**
**(blue squares) in precipitation days (RR1 (days year[-1]), first column, number of days with precipitation amount over 20 mm**
**(R20mm (days year[-1]), second column), simple precipitation intensity index (SDII (mm day[-1]), third column), maximum one day**
**precipitation (Rx1day (mm day[-1]), fourth column) in the Mid-European region (ME). X-axes show the resolution delta (LR/HR)**
**for each model (example: 50 km grid spacing divided by 12.5 km equals 4).**