# Peer review of "The importance of model horizontal resolution on simulated precipitation in Europe – from global to regional models"

_Weather and Climate Dynamics, 2020_

## Short Comment (SC1) · 4 Sep 2020

Comments on "The importance of model resolution on simulated precipitation in Europe – from global to regional model" by Gustav Strandberg and Petter Lind

I would like to make a few comments on this article, which is a big piece of effort, is very interesting and complements a similar analysis by Demory et al. (2020). It's always reassuring to have similar results with different pieces of code and types of analysis.

I would like to point at a few differences between your article and Demory et al. (2020):

- Demory et al. analyse precipitation on a 50km scale (except for CMIP5), whereas you

mix all model resolutions. Klingaman et al. (2017) emphasize that regridding models changes the precipitation distribution as you point out at lines 128. But they argue that models should be compared on similar grids at different scales: a 12km model is meant to be good at 12km, at 50km and at 200km. A 200km model is not meant to be good at 12km. If you use observations only on a 25km scale (as I believe E-OBS is), you cannot expect CMIP5/6 to be good. Similarly, you show that 12km overestimates intense precipitation but this is compared with E-OBS which has a coarser scale than 12km model. In Demory et al., we showed that 12km models overestimated intense precipitation even when regridded at a 50km scale against observation regridded at 50km. Maybe you should include more discussion on this or deserve a few figures to a comparison of everything on a 200km scale, one on a 50km scale.

- You use averaged distributions across grid-points whereas we first pool the data across the region and then plot the distribution. Both methods are equivalent in a flat homogeneous region but not in region with varied topography. You may be smoothing out more the tail of the distribution than we do. Both methods are valid, I'm just highlighting a difference. - We use a new set of bins compared to Klingaman (2017) and Berthou (2018), defined in Berthou et al. (2019) for two reasons: – we wanted pure exponential increase in the bin size so that all the bins have the same size in a log scale and area below the curve is the mean. It's not quite the case in Klingaman and Berthou but it does not make a huge difference. – The other reason was that the Klingaman method had too many bins at the start of the distribution for E-OBS, which does not have a continuous precipitation distribution. I wonder how you managed to have such a smooth distribution for E-OBS, maybe the newer version is improved. Or the spatial averaging of distributions does the job. The equation and the difference between the two sets of bins is shown in Fig. S5 here: https://agupubs.onlinelibrary.wiley.com/action/downloadSupplement?doi=10.1029%2F2019GL083544&file=grl59801-sup-0001-agusuppinfo_revised.pdf

Other comments:

- From your explanation in the method section and the y-axis on the ASoP figures, it seems like you are computing the fractional contribution. This would mean that you care about the shape of the distribution only. However, the figures do show some curves almost always above E-OBS and the integral of the differences is not 0 but >0 (e.g. Fig. 2 SC and ME) : this cannot happen if you normalise each curve by mean precipitation, unless you are normalising all curves by mean precipitation in E-OBS? In Demory et al. 2020, we chose to use actual contributions as we wanted information of both mean and distribution at the same time, to show which bins contribute to mean biases. From your discussion, it seems like you are also discussing actual contributions. Please clarify what you did.

- I agree with the sentence lines 19-21 but I think it applies to models of ~50km: PRIMAVERA-HR, CORDEX-44, CORDEX-11 since you show that CMIP5/6 have very different precipitation distributions and clearly overestimate small intensities. Orographic and coastal regions (AL, FR, IP, MD,) exhibit strong differences (as shown in your Fig. 4). So I would add:

"Once reaching ~50km resolution, the difference between different models is often larger than between the low- and high-resolution versions of the same model, which makes it difficult to quantify the improvement. In this sense the quality of an ensemble is depending more on the models it consists of rather than the average resolution of the ensemble."

- You could also include CMCC in the PRIMAVERA ensemble

- In the accepted version of Demory et al., we consider 45 CORDEX HR and 26 CORDEX LR, so I think sentence line 24-25 is not valid. However, you have other strengths in your study, e.g. comparing the spread between resolution and between models. I think a strong common conclusion of our studies that you highlighted well is that it is best to carefully design an ensemble (across all high-resolution models available (>=50km)) rather than to take an ensemble of opportunity to have a good

representation of precipitation distribution.

- Many of the CMIP6 models have almost not wet days in the IP. Is this a bug or real? In which case it is quite worrying: these models are then very dry in this region.

- You could make use of the E-OBS ensemble rather than just mean in your ASoP figures (although it's already a crowded figure)

References:

Berthou, S., Kendon, E., Rowell, D. P., Roberts, M. J., Tucker, S. O., & Stratton, R. A. (2019). Larger future intensification of rainfall in the West African Sahel in a convection‐permitting model. Geophysical Research Letters, 46, 13299– 13307. https://doi.org/10.1029/2019GL083544

---

## Author Comment (AC1) · 14 Sep 2020

By mistake, old erroneous versions of figures 6-9 were included in the manuscript. The correct versions are included below. This does not affect the conclusions, they were based on correct figures. Thanks to Segolene Berthou for noticing this.
* * *
[Figure]

**Fig. 1.**

**Fig. 2.**

**Fig. 3.**

**Fig. 4.**

[Figure]

---

## Short Comment (SC2) · 25 Sep 2020

**wcd-2020-31: Response to comments by Ségolène Berthou**

Comment #2:
- You use averaged distributions across grid-points whereas we first pool the data across the region and then plot the distribution. Both methods are equivalent in a flat homogeneous region but not in region with varied topography. You may be smoothing out more the tail of the distribution than we do. Both methods are valid, I'm just highlighting a difference. - We use a new set of bins compared to Klingaman (2017) and Berthou (2018), defined in Berthou et al. (2019) for two reasons: – we wanted pure exponential increase in the bin size so that all the bins have the same size in a log scale and area below the curve is the mean. It's not quite the case in Klingaman and Berthou but it does not make a huge difference. – The other reason was that the Klingaman method had too many bins at the start of the distribution for E-OBS, which does not have a continuous precipitation distribution. I wonder how you managed to have such a smooth distribution for E-OBS, maybe the newer version is improved. Or the spatial averaging of distributions does the job. The equation and the difference between the two sets of bins is shown in Fig. S5 here: https://agupubs.onlinelibrary.wiley.com/action/downloadSupplement? doi=10.1029%2F2019GL083544&file=grl59801 sup-0001-agusuppinfo_revised.pdf

> Response: Unfortunately there was an error in the method section describing the ASoP analysis. We actually pooled all grid points across the region prior to ASoP calculations. We have made changes accordingly in the text. An updated version of the section describing ASoP analysis is provided below.
>
> Regarding the bins; we find the arguments for using exponential bin sizes (as used in Berthou et al. 2019) interesting and especially in the case of E-OBS that does not have continuous intensity distribution. In order to increase the readability of the figures, we applied a filter to the resulting distributions to reduce the noise. We've made sure that the smoothed data did not affect the interpretation of the results. However, we failed to include this procedure in the description of ASoP analysis. This has now been corrected for (see text below).

Comment #3

- From your explanation in the method section and the y-axis on the ASoP figures, it seems like you are computing the fractional contribution. This would mean that you care about the shape of the distribution only. However, the figures do show some curves almost always above E-OBS and the integral of the differences is not 0 but >0 (e.g. Fig. 2 SC and ME) : this cannot happen if you normalise each curve by mean precipitation, unless you are normalising all curves by mean precipitation in E-OBS? In Demory et al. 2020, we chose to use actual contributions as we wanted information of both mean and distribution at the same time, to show which bins contribute to mean biases. From your discussion, it seems like you are also discussing actual contributions. Please clarify what you did.

> Response: The labels on the Y-axis were not correct unfortunately. All ASoP figures (except Fig. 4) show actual contributions and not fractional contributions. We have updated the figures and clarified in figure texts what is shown (please see attached figures).

Updated text in Method section, describing ASoP analysis:

[revised manuscript text omitted]

---

## Referee Comment (RC1) · Anonymous Referee #1 · 5 Oct 2020

This study analyses precipitation characteristics over Europe from a wide range of model ensembles, including Global Climate Models (CMIP5, CMIP6, PRIMAVERA) and Regional Climate Models (CORDEX). The precipitation characteristics include daily precipitation distributions based on the ASoP diagnostics developed by Klingaman et al (2017), as well as statistical metrics such as number of wet days, number of heavy precipitation days, intensity of wet days, intensity of heaviest precipitation day. The aim of this study is three-fold: 1) investigate differences between model ensembles, and between models within each ensemble, by using a wide range of ensembles from CMIP5, CMIP6, PRIMAVERA and CORDEX; 2) evaluate model performance against observations, using E-OBS data; 3) investigate the role of resolution in precip-

itation characteristics over Europe, by selecting only models available at both low and high resolution versions.

I have several comments regarding this study, as described below. Some of them would require more analyses and restructuring of the paper, but I think it would also greatly improve it.

1) The authors have made an impressive work by analysing such a huge amount of simulations. This is very complementary to the work by Demory et al (2020), which have analysed daily precipitation over Europe in CMIP5, PRIMAVERA (high-resolution) and CORDEX (low and high resolutions) compared to high-quality observational datasets over Europe. This work has now been revised by focusing more on EUR-11 (which is a newer ensemble than EUR-44), and by including also spatial distribution of precipitation and Taylor diagrams, which confirm the results shown by the precipitation distribution. The paper is now accepted and should appear soon. I suggest to refer to this study already in the introduction. Iles et al (2020) could also be referred to in the introduction as another study evaluating a range of GCMs and RCMs at various resolutions, considering the atmosphere-only UPSCALE simulations. The fact that this study and Demory et al find similar results, despite using slightly different methods, give strength to these two studies and should be discussed further.

2) The authors have managed to combine their results into well-designed figures. However, I feel the 3 goals should not be addressed with the same method. The authors have indeed decided to perform the analyses on the model native grids. This is a good choice for showing what each ensemble is able to simulate at its own resolution, and could be used for addressing aim 1) written above, as long as the models are not compared to each other. A clean comparison could only be done on a common coarser grid, as emphasised by Klingaman et al, 2017. Evaluating results on native grid not only shows the potential of the model physics but also includes the technical aspect of doing analyses on a finer grid. This technical aspect can be evaluated by regridding the data on a coarser grid and see how the results are affected by such a regridding.

Evaluating results on common grids would show the impact of the model physics, its internal resolution solely (Na et al, 2020), and allows a direct assessment and inter-comparison of the results across resolutions (Demory et al, 2020; Iles et al, 2020; see also Torma et al, 2015 (their Fig 3-6)). I would therefore suggest to redo analyses on a common coarser grid to verify the results shown on native grid. I believe this would strengthen the results. One way to answer all 3 aims of the study could be to split it into two parts: the first part would address 1) and 2) on native grids, considering observations available at various resolutions (such as low-resolution satellite data on grids similar to CMIP); the second part would evaluate the impact of resolution by regridding all data on a common coarser grid.

3) The models are evaluated against E-OBS. E-OBS is a good product that tries to gather the highest number of stations currently available. This is particularly the case over Scandinavian regions, or Germany. However, there are still many regions where the station density is low (e.g. France, Italy, Spain, Switzerland, Austria). Over these regions, it would be better to use national gridded datasets, available at much higher resolution (see Demory et al, 2020 for details). I understand the authors may not want to go in that direction, as it adds a lot of processing time and the definitions of the regions would be slightly different than in the current study. I would therefore suggest to include a discussion on this (and eventually an intercomparison with observational results of Demory et al if feasible). Moreover, for aim 1) of the study, I would suggest the authors to use another lower resolution dataset, such as satellite observations, using a resolution closer to CMIP models. This would give an additional range of observational uncertainty.

4) Please verify the use of model resolution when you actually refer to model horizontal grid spacing. The model effective resolution is typically 4 to 8 times the model horizontal grid spacing (Skamarock, 2004; Klaver et al, 2019).

5) Most analyses have been performed annually. It would be good to show them seasonally as well (at least DJF and JJA), as the processes driving precipitation are different and RCMs depend more on GCMs in DJF than JJA (e.g. Hall, 2014; Prein et al, 2016; Fernandez et al, 2019).

6) The abstract needs to be revised. It writes very general conclusions as it stands. See detailed suggestions below. This is true as well for the entire text. Some sentences are a bit hard to read, and in many places, it reads like general statements or approximative sentences. I provided some suggestions for some of them below, but a careful review of the language would clarify the text and be beneficial to the final paper.

7) For reproducibility of the results, it appears important to list the models that were considered for the study.

Detailed comments:

Title: the importance of model 'horizontal' resolution... from global to regional 'models'

L. 10: model 'horizontal' resolution

L. 17-18: I find this conclusion too general. This depends on seasons, and most of the analyses have been performed annually.

L. 20: I don't agree with this. The authors have shown here that the improvement is systematic across models but that there is a large inter-model variability.

L. 21: I agree with this, but I think it cannot be generalised for all resolutions. The authors have shown here that the averaged resolution of CMIP5 and CMIP6 anyway is too low to capture the characteristics of precipitation, at least against E-OBS and other higher resolution ensembles.

L. 22: again, this depends on the season and the authors have mostly worked with annual means.

L. 23: different RCMs driven by the same GCM give different results, but the same RCM driven by different GCMs also give different results (e.g. Vautard et al, 2020).

L. 24-25: Given the complementarity to Demory et al (2020), this sentence needs to be rewritten.

L. 28: delete 'precipitation extremes' in 'precipitation extremes (heavy precipitation events)' -> heavy precipitation events

L. 34: see also Ban et al, 2015

L. 38-39: could the authors add references to support this sentence?

L. 40: 'statistically': remove

L. 40: 'decreasing' -> 'refining'

L. 45: these papers are among many others (e.g. Delworth et al, 2012; Kinter et al, 2013; Roberts et al, 2018 and references therein)

L. 47: please also refer to more recent studies

L. 53-54: Please be careful not to suggest that climate change response in RCM versus GCM may be solely due to resolution. They also depend on the forcings. For example, Boe et al, 2020 and Gutierrez et al, 2020 show the impact of different aerosol treatments between GCM and RCM that may explain part of the different climate change response.

L. 61: please add a reference

L. 62: check the study by Vergara-Temprado et al, 2019. They show that it is possible to turn off convection scheme at such resolution and get appropriate results.

L. 63: 'certain': which ones?

L. 66: 'giving' -> 'simulating'

L. 68: that is true for models with parametrised convection, please also refer to Vergara et al, 2019 (also in L. 71).

L. 77-78: 12km is not high resolution for RCMs, it is its new standard resolution within CORDEX

L. 79: spell out HighResMIP

L. 95: high-resolution PRIMAVERA models are available at higher resolution than 40km at mid-latitude (with is the common referenced latitude), or please specify at which latitude this refers to. I would suggest to use the mid-latitude grid spacing (at 50 degree N), as it is the mid latitude of the European domain (so comparable to EURO-CORDEX grid spacings). It would be clearer to use the term horizontal grid spacing here.

L. 95-97 & Table 1: why not considering the full ensembles? How were the models selected? Why are there 5 PRIMAVERA LR and 4 HR?

Figures 2-3-5: I refer to the revised figures. What does 'act' mean? Please clarify the x-axis 'precipitation bins' and y-axis 'precipitation contribution' labels.

Figures 2-3: specify in the caption that the thick lines are for ensemble means, and that the bottom panels are differences with E-OBS.

Figures 2-3-4: E-OBS is written in Table 1 to be available at 2 resolutions. Which is shown on these figures?

Figures 6-7-8-9-10: I guess E-OBS is shown here at its 2 available resolutions, which one is which?

L. 150: bottom left panel for the Alps. Also, CMIP6 upper end seems to be around 50 mm/day and CORDEX HR over 100mm/day.

Figure 3: The spread is much larger in CORDEX than CMIP6 in JJA. It shows that CORDEX is not so sensitive to the GCM boundary conditions but to different parametrisation schemes in JJA. The spread is determined by the min and max values for both EUR-11 and EUR-44. So are these min and max values only represented by 1 RCM, or 1 RCM-GCM simulation? If the spread is represented by min and max values, wouldn't

it be better to plot the median instead of the mean?

Figure 4: It seems biased to consider EUR-11 as the reference and compare observations to that reference, possibly because, although EUR-11 has a higher resolution, their mean climate seems too wet against high density observations as shown by Demory et al (2020), although I agree observations have undercatch errors. If E-OBS are considered too low resolution and not trustable, considering datasets with higher density stations as the reference would be necessary here. Moreover, the ensembles are clearly compared to each other in this figure, with respect to EUR-11. It would be good to see this analysis performed on a common grid to evaluate how it affects the conclusions. It could be done both at 50km for EUR-11, EUR-44 and PRIMAVERA, and then redone for all datasets at 150 (or even 300km), as done in Torma et al, 2015. Why writing the E-OBS total annual mean in the box if EUR-11 is used as a reference?

L. 181: more strongly biased lower -> more negatively biased: I suggest not to use the word 'bias' when compared to an ensemble, which is itself biased.

L. 193-194: Observations have uncertainties but EUR-11 could also rain too much along coastlines and over topography.

L. 211-212: Fig. 5 shows results for the annual mean, so this conclusion may be different at seasonal means (at least between DJF and JJA), so I would suggest to show these seasonally as well. Moreover, the delta in grid spacing between CORDEX LR (50km) and HR (12.5km) is similar for all models (delta=4), so the impact of resolution is potentially more similar (although it depends on models). This is more complex for the PRIMAVERA models that have various deltas between the LR and HR versions. I counted that deltas vary between 2 for most models, 3 for a couple and 5.4 for the HadGEM3 model (https://www.primavera-h2020.eu/modelling/our-models/). Moreover, note that PRIMAVERA HR uses exactly the same tuning parameters as their LR version, so the effect of resolution solely is seen here (this is not the case for the CORDEX ensembles that may use different model versions). Something that could be interesting

to show here is whether, depending on their delta in grid spacing, some PRIMAVERA models show larger differences than some others. But I would not generalise, based on ensemble means, that resolution in CORDEX has more effect than resolution in PRIMAVERA. It would be good to see the spread of the ensembles on figure 5.

L. 216-218: I agree with this hypothesis, and yet you found greater differences in CORDEX (driven by same low-resolution GCMs) than in PRIMAVERA (L. 211-212). I think this highlights the need for analyses on a common grid, based on seasonal means, and taking into account the fact that CORDEX and PRIMAVERA have different deltas in grid spacing.

L. 226-227: this is not a sentence/question: please rephrase.

Figures 6-9: I considered the revised figures. I still do not understand why some values are not shown. For example: Fig. 6 top left: For one of the CMIP5, only the 10th and 90th percentiles are shown, nothing else it seems. For some other CMIP5 and CMIP6 models, the boxes are drawn but not the whiskers.

Figures 6-7: it seems that CMIP5, CORDEX LR and HR have a larger variability, so is the variability of CORDEX driven by the variability of CMIP5? This could be answered by looking at the seasonal means (DJF and JJA).

L. 227-246: Again for these analyses, the metrics can be analysed for each ensemble on their native resolution, but if the ensembles are compared to each other, as written in the text, then the analyses need to be redone on a common grid.

L. 232-233: Would it be possible to show this with seasonal means?

L. 245: isn't it 20 mm/day instead of 10?

L. 271: rephrase 'negative for some models and positive for some' as it reads too vague

Figure 10: This intercomparison needs to be performed on a common grid

L. 281: left -> right

L. 283: right -> left

L. 283-284: Note that ECMWF HR is 25km grid spacing output at 50km, and LR is 50km output at 100km grid spacing. The delta in grid spacing is therefore 2, and the output are regridded to coarser resolution. This may impact the results.

L. 300-301: Demory et al have revised the manuscript with a focus on EUR-11.

L. 306: give extremes that are heavier and more frequent -> simulate more intense and more frequent heavy precipitation. I would avoid the term 'extremes' with such low-resolution models, and refer instead to 'heavy' or 'intense'.

L. 308: overestimation compared to E-OBS

L. 315: CMIP6 and CMIP5

L. 318-319: this is probably particularly the case for JJA (as shown in fig 3), but for this conclusion it would be good to see DJF and JJA for fig 6-9.

L. 320: not only. E-OBS is not based on the full network of rain gauges over some other countries, such as France.

L. 332: scale -> grid

L. 336: will have -> has

L. 340-341: yes but PRIMAVERA tends to be drier than CORDEX in all seasons.

L. 343: to -> too

L. 344: agree -> agrees

L. 345: the quantification can be done if performed on common grids

L. 348-349: this can depend on seasons

L. 350-351: this needs to be rephrased, as Demory et al have evaluated CMIP5, CORDEX LR/HR and PRIMAVERA HR

Proper acknowledgement needs to be given to the PRIMAVERA, CORDEX, and CMIP modelling groups

There are several typos in the text, please check carefully (e.g. L.8: effects -> affects; L. 20: in depending -> depends; L. 180: region -> regional (and remove comma afterwards); L. 218: were -> where; many others) L. 139 and 141: below/above c: are these typos?

References:

Ban N., Schmidli, J., and Schär, C.: Heavy precipitation in a changing climate: Does short-term summer precipitation increase faster?, Geophys. Res. Lett., 42, 1165–1172, https://doi.org/10.1002/2014GL062588, 2015.

Boé, J., Somot, S., Corre, L., and Nabat, P.: Large differences in Summer climate change over Europe as projected by global and regional climate models: causes and consequences, Clim. Dynam., 54, 2981–3002, https://doi.org/10.1007/s00382-020-05153-1, 2020.

Delworth TL, Rosati A, Anderson W, Adcroft AJ, Balaji V, Benson R, Dixon K, Griffies SM, Lee HC, Pacanowski RC, Vecchi GA, Wittenberg AT, Zeng F, Zhang R (2012) Simulated climate and climate change in the GFDL CM2.5 high-resolution coupled climate model. J Clim 25:2755–2781. doi:10.1175/JCLI-D-11-00316.1

Fernández, J., Frías, M. D., Cabos, W. D., Cofiño, A. S., Domínguez, M., Fita, L., Gaertner, M. A., García-Díez, M., Gutiérrez, J. M., Jiménez-Guerrero, P., Liguori, G., Montávez, J. P., Romera, R., and Sánchez, E.: Consistency of climate change projections from multiple global and regional model intercomparison projects, Clim. Dynam., 52, 1139-1156, https://doi.org/10.1007/s00382-018-4181-8, 2019.

Gutiérrez, C., Somot, S., Nabat, P., Mallet, M., Corre, L., van Meijgaard, E., Perpiñán, O., and Gaertner, M. A.: Future evolution of surface solar radiation and photovoltaic potential in Europe: investigating the role of aerosols, Environ. Res. Lett.,15, 034035,

https://doi.org/10.1088/1748-9326/ab6666, 2020.

Hall, A.: Projecting regional change, Science, 346, 1460–1462, https://doi.org/10.1126/science.aaa0629, 2014.

Kinter III JL, Cash B, Achuthavarier D, Adams J, Altshuler E, Dirmeyer P, Doty B, Huang B, Jin EK, Marx L, Manganello J, Stan C, Wakefield T, Palmer T, Hamrud M, Jung T, Miller M, Towers P, Wedi N, Satoh M, Tomita H, Kodama C, Nasuno T, Oouchi K, Yamada Y, Taniguchi H, Andrews P, Baer T, Ezell M, Halloy C, John D, Loftis B, Mohr R, Wong K (2013) Revolutionizing climate modeling with project Athena: a multi- institutional, international collaboration. Bull Am Meteorol Soc 94:231–245. doi:10.1175/BAMS-D-11-00043.1

Klaver R, Haarsma R,Vidale PL, Hazeleger W. Effective resolution inhigh resolution global atmospheric models forclimate studies. Atmos Sci Lett. 2020;21:e952.

Na, Y., Fu, Q., & Kodama, C. (2020). Precipitation probability and its future changes from a global cloud‐resolving model and CMIP6 simulations. Journal of Geophysical Research: Atmospheres,125, e2019JD031926. https://doi.org/10.1029/2019JD031926

Prein, A. F., Gobiet, A., Truhetz, H., Keuler, K., Goergen, K., Teichmann, C., Fox Maule, C., van Meijgaard, E., Déqué, M., Nikulin, G., Vautard, R., Colette, A., Kjellström, E., and Jacob, D.: Precipitation in the EURO‐CORDEX 0.11° and 0.44° simulations: high resolution, high benefits?, Clim. Dynam., 46, 383-412, https://doi.org/10.1007/s00382-015-2589-y, 2016. Skamarock, W.C., 2004. Evaluating mesoscale NWP models using kinetic energy spectra. Monthly weather review, 132(12), pp.3019-3032.

Roberts, M. J., Vidale, P. L., Senior, C., Hewitt, H. T., Bates, C., Berthou, S., Chang, P., Christensen, H. M., Danilov, S., Demory, M.-E., Griffies, S. M., Haarsma, R., Jung, T., Martin, G., Minobe, S., Ringler, T., Satoh, M., Schiemann, R., Scoccimarro, E.,

[Figure]

Stephens, G., and Wehner, M. F.: The Benefits of Global High Resolution for Climate Simulation: Process Understanding and the Enabling of Stakeholder Decisions at the Regional Scale, B. Am. Meteorol. Soc., 99, 2341–2359, https://doi.org/10.1175/BAMS-D-15-00320.1, 2018.

Torma, C., Giorgi, F., and Coppola, E.: Added value of regional climate modeling over areas characterized by complex terrain—Precipitation over the Alps, J. Geophys. Res. - Atmos., 120, 3957–3972, https://doi.org/10.1002/2014JD022781, 2015.

Vautard, R., Kadygrov, N., Iles, C., Boberg, F., Buonomo, E., Coppola, E., Bülow, K., Corre, L., van Meijgaard, E., Nogherotto, R., et al.: Assessment of the large EURO-CORDEX regional climate simulation ensemble, J. Geophys. Res. – Atmos., in press.

Vergara-Temprado, J., Ban, N., Panosetti, D., Schlemmer, L., and Schär, C. (2019). Climate models permit convection at much coarser resolutions than previously considered. J. Clim., JCLI-D-19-0286.1. doi:10.1175/JCLI-D-19- 0286.1.

---

## Referee Comment (RC2) · Anonymous Referee #2 · 13 Oct 2020

The paper "The importance of model resolution on simulated precipitation in Europe – from global to regional model" by Strandberg and Lind assesses the ability of a large set of climate models in simulating precipitation (particularly extremes) in European sub-regions. The authors find that models with coarse grid spacings underestimate the amount and frequency of extreme precipitation but that the variability between models can be larger than the sensitivity to grid spacing. The novel contribution of this study is the inclusion of global climate model data in their analysis since very similar and more detailed analyses have been done with regional models over Europe. I have two major concerns with this manuscript. First, it does not account for the spatial dependence of extreme precipitation. I argue that the authors can obtain the same

results by first aggregating E-Obs observations to a coarser grid and then comparing the aggregated extreme precipitation with the original E-Obs data. They would also see that the coarser version of E-Obs "underestimates" extreme frequency and magnitude. Coarse-resolution models should not reproduce the magnitude of extreme events on local scales since they model aggregated rainfall over large areas (e.g., 100x100 km). My second concern is the use of E-Obs for this analysis. E-Obs has very low station density over large parts of Europe and heavily underestimates extreme precipitation. There are other observational datasets available that are far more appropriate for the presented analysis. More details on these comments including relevant literature is provided below.

General Comments: 1. I have major concerns with your approach to compare extreme precipitation. Extreme precipitation is strongly scale dependent and largest on point scales (e.g. measured by precipitation gauges) and decreases on larger spatial-scales. E-OBS for example has way weaker extreme precipitation than other regional datasets in Europe that feature higher resolution and a higher station density (e.g. Prein and Gobiet 2017). If you compare extreme precipitation on the model native grid, you mix the model ability in simulating extreme precipitation with the spatial scale on which the model simulates extremes. E.g., extreme precipitation in a 100 km grid spacing model should not match observed extreme precipitation on a 25 km grid. In this case the only way to do a fair comparison is to aggregate the 25 km grid observations to the 100 km model grid. This aggregation does not introduce large biases such as you state for interpolation (in Line 127-128).

2. E-Obs should be used with care for extreme precipitation (Haylock et al. 2008). There are other/regional datasets in Europe that are much better suited for the assessment of extreme precipitation (see Prein and Gobiet et al. 2017).

3. You are missing to discuss and to refer relevant literature on the ERUO-CORDEX simulations that performed very similar analysis as you present. Kotlarski et al. (2014), Casanueva et al. (2016), and Prein et al. (2016) address similar questions and come

to fairly similar conclusions. The novelty of your analysis is that you also include GCM data, which is a valuable contribution but does not change the major conclusions. You should also take a look at Thackeray et al. (2018) who show a highly relevant analysis of model grid spacing and extreme precipitation on a global-scale.

4. Please be careful with the use of model resolution. In most cases you refer to model grid spacing. Model resolution depends on the numeric diffusion in the model and models with the same grid spacing can have different resolutions. The effective resolution of a model is typically 4-8 times its grid spacing (e.g., Skamarock 2004).

5. There are many typos and grammar errors in the document. Please consider using a proofreader before resubmitting the document.

Literature: Casanueva, A., Kotlarski, S., Herrera, S., Fernández, J., Gutiérrez, J.M., Boberg, F., Colette, A., Christensen, O.B., Goergen, K., Jacob, D. and Keuler, K., 2016. Daily precipitation statistics in a EURO-CORDEX RCM ensemble: added value of raw and bias-corrected high-resolution simulations. Climate dynamics, 47(3-4), pp.719-737.

Haylock, M.R., Hofstra, N., Klein Tank, A.M.G., Klok, E.J., Jones, P.D. and New, M., 2008. A European daily high‐resolution gridded data set of surface temperature and precipitation for 1950–2006. Journal of Geophysical Research: Atmospheres, 113(D20).

Kotlarski, S., Keuler, K., Christensen, O.B., Colette, A., Déqué, M., Gobiet, A., Goergen, K., Jacob, D., Lüthi, D., Van Meijgaard, E. and Nikulin, G., 2014. Regional climate modeling on European scales: a joint standard evaluation of the EURO-CORDEX RCM ensemble. Geoscientific Model Development, 7, pp.1297-1333.

Prein, A.F. and Gobiet, A., 2017. Impacts of uncertainties in European gridded precipitation observations on regional climate analysis. International Journal of Climatology, 37(1), pp.305-327.

Prein, A.F., Gobiet, A., Truhetz, H., Keuler, K., Goergen, K., Teichmann, C., Maule, C.F., Van Meijgaard, E., Déqué, M., Nikulin, G. and Vautard, R., 2016. Precipitation in the EURO-CORDEX 0.11 and 0.44âĹ̆Ÿ simulations: high resolution, high benefits?. Climate dynamics, 46(1-2), pp.383-412.

Skamarock, W.C., 2004. Evaluating mesoscale NWP models using kinetic energy spectra. Monthly weather review, 132(12), pp.3019-3032.

Thackeray, C.W., DeAngelis, A.M., Hall, A., Swain, D.L. and Qu, X., 2018. On the connection between global hydrologic sensitivity and regional wet extremes. Geophysical Research Letters, 45(20), pp.11-343.

---

## Editor Comment (EC1) · Martin Singh (Editor) · 14 Oct 2020

The manuscript has now received two solicited reviews and a short comment, and the interactive discussion period is closed. The reviews are encouraging, but they all note a few shortcomings of the methods that should be addressed in a revised submission. In particular, the authors should address concerns relating to their comparison of precipitation distributions in models with different grid spacing and the appropriateness of the observational dataset used.

I encourage the authors to submit a revised manuscript that satisfactorily addresses all comments.

---

## Author Comment (AC2) · 24 Nov 2020

**From Ségolène Berthou:**
I would like to make a few comments on this article, which is a big piece of effort, is very interesting and complements a similar analysis by Demory et al. (2020). It's always reassuring to have similar results with different pieces of code and types of analysis. I would like to point at a few differences between your article and Demory et al. (2020):

We thank Ségolène Berthou for making the effort of reviewing, and for all valuable comments. Responses follow in red below. In the markup version of the revised manuscript substantial changes are also marked in red.

- Demory et al. analyse precipitation on a 50km scale (except for CMIP5), whereas you mix all model resolutions. Klingaman et al. (2017) emphasize that regridding models changes the precipitation distribution as you point out at lines 128. But they argue that models should be compared on similar grids at different scales: a 12km model is meant to be good at 12km, at 50km and at 200km. A 200km model is not meant to be good at 12km. If you use observations only on a 25km scale (as I believe E-OBS is), you cannot expect CMIP5/6 to be good. Similarly, you show that 12km overestimates intense precipitation but this is compared with E-OBS which has a coarser scale than 12km model. In Demory et al., we showed that 12km models overestimated intense precipitation even when regridded at a 50km scale against observation regridded at 50km. Maybe you should include more discussion on this or deserve a few figures to a comparison of everything on a 200km scale, one on a 50km scale.

We see what you mean; it is, of course, in a sense unfair to compare models of different resolutions. We assume that models of higher resolution will perform better than models of lower resolution; and a model on 12 km will be extra good if the observations are also on 12 km. On the other hand, when you are about to use data from climate models the choice is for example between GCM and RCM, or between RCM of low resolution and RCM of high resolution. Or perhaps you are thinking about if it's worth the effort of making atmosphere only GCM runs to increase the resolution instead of just using standard GCM results. Then you will use the data of choice and perhaps compare it to observations, other models etc. Therefore we made the active choice of using this method because it allows us to preserve the model output on its native grid.
Nevertheless, we see the need of also comparing at common grids. We have now included analyses when all data are regridded to a 0.5°×0.5° grid and a 2°×2°grid.

- You use averaged distributions across grid-points whereas we first pool the data across the region and then plot the distribution. Both methods are equivalent in a flat homogeneous region but not in region with varied topography. You may be smoothing out more the tail of the distribution than we do. Both methods are valid, I'm just highlighting a difference.
- We use a new set of bins compared to Klingaman (2017) and Berthou (2018), defined in Berthou et al. (2019) for two reasons: – we wanted pure exponential increase in the bin size so that all the bins have the same size in a log scale and area below the curve is the mean. It's not quite the case in Klingaman and Berthou but it does not make a huge difference. – The other reason was that the Klingaman

method had too many bins at the start of the distribution for E-OBS, which does not have a continuous precipitation distribution. I wonder how you managed to have such a smooth distribution for E-OBS, maybe the newer version is improved. Or the spatial averaging of distributions does the job. The equation and the difference between the two sets of bins is shown in Fig. S5 here:
https://agupubs.onlinelibrary.wiley.com/action/downloadSupplement?doi=10.1029%2F2019GL083544&file=grl59801-sup-0001-agusuppinfo_revised.pdf

Unfortunately there was an error in the method section describing the ASoP analysis. We actually pooled all grid points across the region prior to ASoP calculations. We have made changes accordingly in the text. An updated version of the section describing ASoP analysis is provided below.
Regarding the bins; we find the arguments for using exponential bin sizes (as used in Berthou et al. 2019) interesting and especially in the case of E-OBS that does not have continuous intensity distribution. In order to increase the readability of the figures, we applied a filter to the resulting distributions to reduce the noise. We've made sure that the smoothed data did not affect the interpretation of the results. However, we failed to include this procedure in the description of ASoP analysis. This has now been corrected for (see text below).

Other comments:
- From your explanation in the method section and the y-axis on the ASoP figures, it seems like you are computing the fractional contribution. This would mean that you care about the shape of the distribution only. However, the figures do show some curves almost always above E-OBS and the integral of the differences is not 0 but >0 (e.g. Fig. 2 SC and ME): this cannot happen if you normalise each curve by mean precipitation, unless you are normalising all curves by mean precipitation in E-OBS? In Demory et al. 2020, we chose to use actual contributions as we wanted information of both mean and distribution at the same time, to show which bins contribute to mean biases. From your discussion, it seems like you are also discussing actual contributions. Please clarify what you did.

The labels on the Y-axis were not correct unfortunately. All ASoP figures (except Fig. 4) show actual contributions and not fractional contributions. We have updated the figures and clarified in figure texts what is shown (please see attached figures).

Updated text in Method section, describing ASoP analysis:
"To investigate the effect of model grid resolution on the full distributions of daily precipitation intensities, we use the ASoP (Analysing Scales of Precipitation) method (Klingaman et al., 2017; Berthou et al., 2018). ASoP involves splitting precipitation distributions into bins of different intensities and then provides information of the contribution from each precipitation intensity separately to the total mean precipitation rate (i.e. given by all intensities taken together). In the first step, precipitation intensities are binned in such a way that each bin contains a similar number of events, with the exception of most intense events, which are rare. The actual contribution (in mm) of each bin to the total mean precipitation rate is obtained by multiplying the frequency of events by the mean precipitation rate. The sum of the actual contributions from all bins gives the total mean precipitation rate. The fractional contribution (in %) of each bin is further obtained by dividing the actual contributions by the mean precipitation rate. In this case, the sum of all fractional contributions is

equal to one, thus the information provided by fractional contributions is predominantly about the shape of the distribution. Taking the absolute differences between two fractional distributions and sum over all bins gives a measure of the difference in the shapes of the precipitation distributions. This is here called the "Index of fractional contributions". Since E-OBS precipitation intensities, in contrast to model data, are not continuous the resulting ASoP factors for E-OBS tend to be noisy, especially for lower intensities. In order to facilitate the interpretation of the results, the regionally averaged ASoP factors for E-OBS were smoothed to some extent by using a simple filter.
The ASoP method is here applied to grid points pooled over target regions (Fig. 1) separately and the result is a distribution for each model showing the probability of different precipitation intensities based on daily precipitation. Most results presented here concern the actual contributions, both to limit the number of figures and because these factors conveniently provide information on both shape of distributions as well as the mean values. The ASoP distributions of all analysed models are used to compare model behaviour and performance. In particular to see how changing the grid resolution affects different parts of the distribution, for example if contributions from low and high precipitation intensities are different."

- I agree with the sentence lines 19-21 but I think it applies to models of ~50km: PRIMAVERA-HR, CORDEX-44, CORDEX-11 since you show that CMIP5/6 have very different precipitation distributions and clearly overestimate small intensities. Orographic and coastal regions (AL, FR, IP, MD,) exhibit strong differences (as shown in your Fig. 4). So I would add:
"Once reaching ~50km resolution, the difference between different models is often larger than between the low- and high-resolution versions of the same model, which makes it difficult to quantify the improvement. In this sense the quality of an ensemble is depending more on the models it consists of rather than the average resolution of the ensemble."
We change the sentence accordingly.

-You could also include CMCC in the PRIMAVERA ensemble

We tried to get daily pr data of CMCC from the CEDA archive, but didn't manage to get it.

- In the accepted version of Demory et al., we consider 45 CORDEX HR and 26 CORDEX LR, so I think sentence line 24-25 is not valid. However, you have other strengths in your study, e.g. comparing the spread between resolution and between models. I think a strong common conclusion of our studies that you highlighted well is that it is best to carefully design an ensemble (across all high-resolution models available (>=50km)) rather than to take an ensemble of opportunity to have a good representation of precipitation distribution.
You're right. That was perhaps a bit exaggerated. We change the sentence to:
The results presented here are in line with previous similar studies. To these studies we add details about the spread between resolutions and between models.

- Many of the CMIP6 models have almost not wet days in the IP. Is this a bug or real? In which case it is quite worrying: these models are then very dry in this region.

This is a bug. Wrong versions of figures 6-9 were accidently inserted in the manuscript. This is now corrected.

- You could make use of the E-OBS ensemble rather than just mean in your ASoP figures (although it's already a crowded figure)
Individual E-OBS members are available upon request, but as we understand it these are useful if you want to sample uncertainty when you use E-OBS as forcing. E-OBS writes: "The individual ensemble members are mainly intended for users who require the uncertainty in the gridded fields to propagate through to various other applications. ..."
If we were looking at specific events this could perhaps be interesting, but since we look at climatologies we don't see the use of crowding this figure even more.

References:
Berthou, S., Kendon, E., Rowell, D. P., Roberts, M. J., Tucker, S. O., & Stratton, R. A. (2019). Larger future intensification of rainfall in the West African Sahel in a convection˘AR˘ permitting model. Geophysical Research Letters, 46, 13299–13307. https://doi.org/10.1029/2019GL083544

---

## Author Comment (AC3) · 24 Nov 2020

The comment was uploaded in the form of a supplement:
https://wcd.copernicus.org/preprints/wcd-2020-31/wcd-2020-31-AC3-supplement.pdf

---

## Author Comment (AC4) · 24 Nov 2020

This study analyses precipitation characteristics over Europe from a wide range of model ensembles, including Global Climate Models (CMIP5, CMIP6, PRIMAVERA)and Regional Climate Models (CORDEX). The precipitation characteristics include daily precipitation distributions based on the ASoP diagnostics developed by Klingaman et al (2017), as well as statistical metrics such as number of wet days, number of heavy precipitation days, intensity of wet days, intensity of heaviest precipitation day. The aim of this study is three-fold: 1) investigate differences between model ensembles, and between models within each ensemble, by using a wide range of ensembles from CMIP5, CMIP6, PRIMAVERA and CORDEX; 2) evaluate model performance against observations, using E-OBS data; 3) investigate the role of resolution in precipitation characteristics over Europe, by selecting only models available at both low and high resolution versions. I have several comments regarding this study, as described below. Some of them would require more analyses and restructuring of the paper, but I think it would also greatly improve it.

We thank you for making the effort of reviewing the paper and for all constructive suggestions. Responses to comments follow below in red. In the markup version of the revised manuscript substantial changes are also marked in red.

1) The authors have made an impressive work by analysing such a huge amount of simulations. This is very complementary to the work by Demory et al (2020), which have analysed daily precipitation over Europe in CMIP5, PRIMAVERA (high-resolution) and CORDEX (low and high resolutions) compared to high-quality observational datasets over Europe. This work has now been revised by focusing more on EUR-11 (which is a newer ensemble than EUR-44), and by including also spatial distribution of precipitation and Taylor diagrams, which confirm the results shown by the precipitation distribution. The paper is now accepted and should appear soon. I suggest to refer to this study already in the introduction. Iles et al (2020) could also be referred to in the introduction as another study evaluating a range of GCMs and RCMs at various resolutions, considering the atmosphere-only UPSCALE simulations. The fact that this study and Demory et al find similar results, despite using slightly different methods, give strength to these two studies and should be discussed further.

The Introduction has been expanded with a paragraph discussing Demory et al., and Iles et al., as well as other similar studies using CORDEX data:

A few studies have been made investigating how model resolution affects the simulated precipitation in the CORDEX ensembles, comparing 50 km and 12.5 km grid spacing. A clear result is that precipitation generally increases with higher resolution, which sometimes means that the bias increases when precipitation is added to already wet models (Kotlarski et al., 2014; Casanueva et al., 2016); something that is also seen in simulations with global models (e.g. Thackeray et al., 2018). An overall improvement of mean precipitation is not seen the high resolution CORDEX simulations, except for regions with complex topography (Korlarski et al., 2014; Casanueva et al., 2016; Prein et al., 2016). Prein et al. (2016) looked at local precipitation on short time scales. They find that 12.5 km simulations better represents extreme and mean precipitation, also when simulations are aggregated to 50 km. They note,

however, that the results are highly dependent on which observations the simulations are compared with. They also note that improvements are on the ensemble as a whole, and not necessarily for each individual model. In similar studies as the present Iles et al. (2019) and Demory et al. (2020) compare CORDEX simulations with simulations from CMIP5 and Primavera. They see that precipitation increases with resolution so that CMIP5 underestimates precipitation amounts and CORDEX overestimates it, when compared to E-OBS, and that the effect of resolution is largest in complex topography. They also find that Primavera performs similarly to CORDEX when run on the same resolution, which is interesting regarding that the Primavera models are developed for low resolution. Iles et al. (2019) also find considerable inter-model differences meaning that improvements are seen on the ensemble level rather that for individual models.

2) The authors have managed to combine their results into well-designed figures. However, I feel the 3 goals should not be addressed with the same method. The authors have indeed decided to perform the analyses on the model native grids. This is a good choice for showing what each ensemble is able to simulate at its own resolution, and could be used for addressing aim 1) written above, as long as the models are not com-pared to each other. A clean comparison could only be done on a common coarser grid, as emphasised by Klingaman et al, 2017. Evaluating results on native grid not only shows the potential of the model physics but also includes the technical aspect of doing analyses on a finer grid. This technical aspect can be evaluated by regridding the data on a coarser grid and see how the results are affected by such a regridding. Evaluating results on common grids would show the impact of the model physics, its internal resolution solely (Na et al, 2020), and allows a direct assessment and inter-comparison of the results across resolutions (Demory et al, 2020; Iles et al, 2020; see also Torma et al, 2015 (their Fig 3-6)). I would therefore suggest to redo analyses on a common coarser grid to verify the results shown on native grid. I believe this would strengthen the results. One way to answer all 3 aims of the study could be to split it into two parts: the first part would address 1) and 2) on native grids, considering observations available at various resolutions (such as low-resolution satellite data on grids similar to CMIP); the second part would evaluate the impact of resolution by regridding all data on a common coarser grid.

Thanks for pushing us in this direction. We have now included analyses where all data are regridded to two common grids 0.5°×0.5° grid and a 2°×2°grid.

3) The models are evaluated against E-OBS. E-OBS is a good product that tries to gather the highest number of stations currently available. This is particularly the case over Scandinavian regions, or Germany. However, there are still many regions where the station density is low (e.g. France, Italy, Spain, Switzerland, Austria). Over these regions, it would be better to use national gridded datasets, available at much higher resolution (see Demory et al, 2020 for details). I understand the authors may not want to go in that direction, as it adds a lot of processing time and the definitions of the regions would be slightly different than in the current study. I would therefore suggest to include a discussion on this (and eventually an intercomparison with observational results of Demory et al if feasible). Moreover, for aim 1) of the study, I would suggest the authors to use another lower resolution dataset, such as satellite observations, using a resolution closer to CMIP models. This would give an additional range of observational uncertainty.

Thank you, this is indeed an important and interesting issue. We agree that it would be valuable if regional and/or national observational datasets (with assumed higher quality than E-OBS) could be included for each of the investigated sub-regions, as for example in Demory et al 2020. As suggested we have now included a separate section with a discussion of observations and their associated uncertainties, including E-OBS. To emphasize the importance of high-quality observations and to partly put our results into perspective, we have also included an ASoP analysis comparing to another high-resolution (1x1 km) dataset covering Scandinavia - called NGCD (Nordic Gridded Climate Dataset). There we can clearly see the impact of including such observations, increasing the confidence in the high-resolution RCM model ensemble. We have not included any satellite data as these often (at least the ones we are aware of) has limited coverage or lower quality over high latitudes.

4) Please verify the use of model resolution when you actually refer to model horizontal grid spacing. The model effective resolution is typically 4 to 8 times the model horizontal grid spacing (Skamarock, 2004; Klaver et al, 2019).

Thanks for reminding us about this. We tried to straighten up the terminology so that we use "grid spacing" when talking about distances in km and "resolution" in more general statements, like comparing high and low resolution models.

5) Most analyses have been performed annually. It would be good to show them seasonally as well (at least DJF and JJA), as the processes driving precipitation are different and RCMs depend more on GCMs in DJF than JJA (e.g. Hall, 2014; Prein et al,2016; Fernandez et al, 2019).

We have now also included analyses of DJF and JJA. However we could not present results for all regions, seasons and resolutions as this would mean at least a 12 fold increase of the number of figures.

6) The abstract needs to be revised. It writes very general conclusions as it stands. See detailed suggestions below. This is true as well for the entire text. Some sentences area bit hard to read, and in many places, it reads like general statements or approximative sentences. I provided some suggestions for some of them below, but a careful review of the language would clarify the text and be beneficial to the final paper.

The abstract is rewritten to be more precise, and so is the rest of the text. We hope in a satisfactory way. Thanks for the detailed comments.

7) For reproducibility of the results, it appears important to list the models that were considered for the study.

We have inserted a new Table 1 listing the GCMs and a new Table 2 listing the RCMs.

Detailed comments:

Title: the importance of model 'horizontal' resolution... from global to regional 'models'

Changed as suggested.

L. 10: model 'horizontal' resolution

Changed as suggested.

L. 17-18: I find this conclusion too general. This depends on seasons, and most of the analyses have been performed annually.

The abstract is rewritten to be more precise.

L. 20: I don't agree with this. The authors have shown here that the improvement is systematic across models but that there is a large inter-model variability.

This is rephrased to: "Even though higher resolution improves the simulated precipitation in a systematic way, the inter-model variability is still large. This means that the quality of an ensemble depends also on the models it consists of and not only the average resolution of the ensemble."

L. 21: I agree with this, but I think it cannot be generalised for all resolutions. The authors have shown here that the averaged resolution of CMIP5 and CMIP6 anyway is too low to capture the characteristics of precipitation, at least against E-OBS and other higher resolution ensembles.

We imply that this is valid for the resolutions used in RCMs. To make this clearer we start the section with "Once reaching ~50 km …".

L. 22: again, this depends on the season and the authors have mostly worked with annual means.

The abstract is rewritten to be more precise.

L. 23: different RCMs driven by the same GCM give different results, but the same RCM driven by different GCMs also give different results (e.g. Vautard et al, 2020).

That's true, and we know this of course. We show it in Fig. 10 and mention it at a few different times. We change the sentence to : "The result of a RCM simulation depends on the driving GCM, but the difference in simulated precipitation between an RCM and the driving GCM depends more on the choice of RCM and less on the down-scaling itself; as different RCMs driven by the same GCM may give different results."

If Vautard et al., 2020 is published before this goes to print we will add a reference to that.

L. 24-25: Given the complementarity to Demory et al (2020), this sentence needs to be rewritten.

This is changed to: "The results presented here are in line with previous similar studies. To these studies we add details about the spread between resolutions and between models."

L. 28: delete 'precipitation extremes' in 'precipitation extremes (heavy precipitation events)' -> heavy precipitation events

Changed as suggested

L. 34: see also Ban et al, 2015

A reference to Ban et al., 2015 is added.

L. 38-39: could the authors add references to support this sentence?

We added references to Champion et al., 2011; Zappa et al., 2013.

L. 40: 'statistically': remove

Changed as suggested

L. 40: 'decreasing' -> 'refining'

Changed as suggested

L. 45: these papers are among many others (e.g. Delworth et al, 2012; Kinter et al,2013; Roberts et al, 2018 and references therein)

We added these references.

L. 47: please also refer to more recent studies

We added references to Dai 2006; Stratton and Stirling, 2012; Gao et al., 2017

L. 53-54: Please be careful not to suggest that climate change response in RCM versus GCM may be solely due to resolution. They also depend on the forcings. For example, Boe et al, 2020 and Gutierrez et al, 2020 show the impact of different aerosol treatments between GCM and RCM that may explain part of the different climate change response.

Thanks for pointing this out. We added the sentence: "Differences in the treatment of aeorosols are also identified as a reason for differences is climate response between RCMs and GCMs (Boé et al., 2020; Gutiérrez et al., 2020)."

L. 61: please add a reference

We added a reference to Iorio et al., 2004

L. 62: check the study by Vergara-Temprado et al, 2019. They show that it is possible to turn off convection scheme at such resolution and get appropriate results.

These are interesting results, but don't change the fact that most simulations on 10 km parameterize convection. We changed the sentence to: "Even at grid spacings of around 10 km convection is usually not resolved by the model dynamics but is instead parameterized (although it might be possible to turn off the parameterization already at this kind of resolution (Vergara-Temprado et al., 2019))."

L. 63: 'certain': which ones?

Mainly the diurnal cycle. We changed the sentence to: "However, models with parameterized convection often exhibit common biases in the diurnal precipitation cycle"

L. 66: 'giving' -> 'simulating'

Changed as suggested

L. 68: that is true for models with parametrised convection, please also refer to Vergara et al, 2019 (also in L. 71).

We change the sentence to: "A deficiency of parameterized convection is that it starts too early (e.g. Dai and Trenberth, 2004; Dai, 2006; Brockhaus et al., 2008; Vergara-Temprado et al., 2019)." And also add a reference to Vergada-Temprado et al., 2019 on L.71.

L. 77-78: 12km is not high resolution for RCMs, it is its new standard resolution within CORDEX

What we refer to here are simulations with "convective permitting resolution" which is <5 km (e.g. Coppola et al., 2018)

L. 79: spell out HighResMIP

Changed as suggested

L. 95: high-resolution PRIMAVERA models are available at higher resolution than 40km at mid-latitude (with is the common referenced latitude),  or please specify at which latitude this refers to. I would suggest to use the mid-latitude grid spacing (at 50 degreeN), as it is the mid latitude of the European domain (so comparable to EURO-CORDEX grid spacings). It would be clearer to use the term horizontal grid spacing here.

This was not so much a matter of latitudes, but a writing mistake. Never the less, it's a good suggestion to spell out mid-latitude grid spacing. We change as suggested "The models used in this study are a selection of CMIP5 global models (~100-300 km mid-latitude horizontal grid spacing); the high (~25-50 km mid-latitude) and low (~80-160 km mid latitude) resolution versions of the PRIMAVERA global models and the first models from CMIP6 (~100-300 km); and a selection of CORDEX regional models (at 12.5 and 50 km mid-latitude grid spacing)."

L. 95-97 & Table 1:  why not considering the full ensembles?  How were the models selected? Why are there 5 PRIMAVERA LR and 4 HR?

We selected the models for which we at the time could get daily precipitation. Since we thought that we got ensembles of reasonable sizes we decided not to track down individual models that were not available in common storages. The Primavera LR and HR ensembles are of different resolutions because HadGEM3-GC31 was run at three resolutions. Only one (25 km) was considered as HR, the other two (60 & 130 km) were considered as LR.

Figures 2-3-5: I refer to the revised figures.  What does 'act' mean?  Please clarify the x-axis 'precipitation bins' and y-axis 'precipitation contribution' labels.

We have updated these figures, and hopefully the titles and axis annotations are more clear now. The figure labels have also been updated to more clearly describe the figure contents.

Figures 2-3: specify in the caption that the thick lines are for ensemble means, and that the bottom panels are differences with E-OBS.

Thanks. The figure labels have been updated accordingly.

Figures 2-3-4:  E-OBS is written in Table 1 to be available at 2 resolutions.  Which is shown on these figures?

In these figures we use E-OBS with the highest resolution (0.1 deg). It is now specified in the figure labels.

Figures 6-7-8-9-10: I guess E-OBS is shown here at its 2 available resolutions, which one is which?

Correct. We added: "E-OBS at 0.25° (grey) and 0.1° km (black)."

L. 150: bottom left panel for the Alps. Also, CMIP6 upper end seems to be around 50mm/day and CORDEX HR over 100mm/day.

Correct, bottom left and bottom right was mixed up. The sentence is changed

Figure 3:  The spread is much larger in CORDEX than CMIP6 in JJA. It shows that CORDEX is not so sensitive to the GCM boundary conditions but to different parametrisation schemes in JJA. The spread is determined by the min and max values for bothEUR-11 and EUR-44. So are these min and max values only represented by 1 RCM, or1 RCM-GCM simulation? If the spread is represented by min and max values, wouldn't it be better to plot the median instead of the mean?

Indeed, the spread defined by max/min values is very sensitive to possible "outliers" that might not be a good representation of the ensemble spread. It is not entirely clear what the best way would be to indicate the spread of such relatively small ensembles (without the use of more sophisticated statistical techniques like bootstrapping). We have changed from max/min to instead show the 5-95 percentile range. We further agree that median values would be more appropriate than mean values and thus have changed accordingly.

Figure 4:  It seems biased to consider EUR-11 as the reference and compare observations to that reference, possibly because, although EUR-11 has a higher resolution, their mean climate seems too wet against high density observations as shown by Demory et al (2020), although I agree observations have undercatch errors.  If E-OBS are considered too low resolution and not trustable, considering datasets with higher density stations as the reference would be necessary here.  Moreover, the ensembles are clearly compared to each other in this figure, with respect to EUR-11.  It would be good to see this analysis performed on a common grid to evaluate how it affects the conclusions.  It could be done both at 50km for EUR-11, EUR-44 and PRIMAVERA, and then redone for all datasets at 150 (or even 300km), as done in Torma et al, 2015.Why writing the E-OBS total annual mean in the box if EUR-11 is used as a reference?

We have now included analysis on common grids (at two different resolutions, 0.5°x0.5° and 2°x2°), although not presented in the format as shown in Fig. 4 (see Figures S1 and S2 in Supplementary). The interpolation to common grids of course have an effect but the overall conclusions are not seriously impacted. Further on, as mentioned above, we included another, regional high-quality, data set in an ASoP analysis to emphasize the importance of such data sets and possible impact on the results. Still, we are limited for most regions to the E-OBS data as reference while acknowledging its inherent uncertainties. Regarding Fig.4 your concerns about having EUR-11 as reference is understandable and we have changed to E-OBS as reference instead.

L. 181: more strongly biased lower -> more negatively biased: I suggest not to use the word 'bias' when compared to an ensemble, which is itself biased.

We changed to: "Region total seasonal precipitation (averaged within each ensemble), are either mostly in the range of +/- 20 % from CORDEX HR (e.g. eastern Europe, EA) or with larger negative values…".

L. 193-194: Observations have uncertainties but EUR-11 could also rain too much along coastlines and over topography.

True, we changed to "…both factors contributing to uncertainties in quality and representativeness of observational and simulated data."

L. 211-212: Fig. 5 shows results for the annual mean, so this conclusion may be different at seasonal means (at least between DJF and JJA), so I would suggest to show these seasonally as well. Moreover, the delta in grid spacing between CORDEX LR(50km) and HR (12.5km) is similar for all models (delta=4), so the impact of resolution is potentially more similar (although it depends on models). This is more complex for the PRIMAVERA models that have various deltas between the LR and HR versions. I counted that deltas vary between 2 for most models, 3 for a couple and 5.4 for theHadGEM3 model (https://www.primavera-h2020.eu/modelling/our-models/).

Moreover, note that PRIMAVERA HR uses exactly the same tuning parameters as their LR version, so the effect of resolution solely is seen here (this is not the case for the CORDEX ensembles that may use different model versions). Something that could be interesting to show here is whether, depending on their delta in grid spacing, some PRIMAVERA models show larger differences than some others. But I would not generalise, based on ensemble means, that resolution in CORDEX has more effect than resolution in PRIMAVERA. It would be good to see the spread of the ensembles on figure 5.

A good point. We added the sentences: "Some differences between the CORDEX and PRIMAVERA ensembles should be noted. The PRIMAVERA models use the same tuning parameters for both the LR and HR version, but on the other hand the differences in resolution between LR and HR varies between models. The CORDEX ensembles have the same difference in resolution for all models, but the LR and HR simulations may be run with different models versions. Hence, all differences between PRIMAVERA and CORDEX ensembles can't be generalised to be attributed by resolution alone."

We also plotted the absolute difference in the precipitation indices between LR and HR against the ratio LR/HR. It turns out the the correlation is weak, e.g. the spread within CORDEX ensemble is large although all models have the same ratio.

In Figure 5 the absolute values for each model have now been included as well (in addition to the ensemble means) showing the ensemble spread.

L. 216-218: I agree with this hypothesis, and yet you found greater differences in CORDEX (driven by same low-resolution GCMs) than in PRIMAVERA (L. 211-212).I think this highlights the need for analyses on a common grid, based on seasonal means, and taking into account the fact that CORDEX and PRIMAVERA have different deltas in grid spacing.

A description of winter and summer is included in the text. Our analysis on common grids and of resolution delta doesn't suggest that this explains the differences. Rather, the conclusion is that for high intensities model resolution and performance is more important

than the driving GCM. We don't know the full answer. This section was also meant to show that there are unresolved issues and to point to possible future studies.

We added the following:"Still, the largest differences are seen in the CORDEX ensemble where the LR and HR models are run with the same coarse resolution GCM. This suggests that (regional) model resolution and performance is what determines high precipitation rates, rather than the driving GCM. "

L. 226-227: this is not a sentence/question: please rephrase.

We changed to: "When do intense precipitation events occur in the high-resolution models? the kind of events that are rarely seen or absent in the low resolutions simulations."

Figures 6-9: I considered the revised figures. I still do not understand why some values are not shown.  For example: Fig.  6 top left: For one of the CMIP5, only the 10th and90th percentiles are shown, nothing else it seems. For some other CMIP5 and CMIP6models, the boxes are drawn but not the whiskers.

In small regions like the Alps and in models of coarse resolution the number of data points are actually too few to make good statistics. This means that calculation of percentiles can be difficult. Since this only happens in some regions for a small number of models we consider this a major problem.

Figures 6-7: it seems that CMIP5, CORDEX LR and HR have a larger variability, so is the variability of CORDEX driven by the variability of CMIP5? This could be answered by looking at the seasonal means (DJF and JJA).

We don't agree that the variability is large in CMIP5, rather the variability increases with resolution. The signal is the same for the individual seasons, but less pronounced since the potential number of days is smaller when divided over four seasons instead of counted over the whole year.

L. 227-246: Again for these analyses, the metrics can be analysed for each ensemble on their native resolution, but if the ensembles are compared to each other, as written in the text, then the analyses need to be redone on a common grid.

Yes, descriptions of summer and winter are now included and figures of this when relevant. The analysis now also include data on common grids.

L. 232-233: Would it be possible to show this with seasonal means?

Yes, descriptions of summer and winter are now included and figures of this when relevant.

L. 245: isn't it 20 mm/day instead of 10?

Yes, we changed to 20.

L. 271: rephrase 'negative for some models and positive for some' as it reads too vague

We changed to: "The differences are small, mainly within ±10 days year$^{-1}$."

Figure 10: This intercomparison needs to be performed on a common grid

This is now done.

L. 281: left -> right

Changed as suggested

L. 283: right -> left

Changed as suggested

L. 283-284: Note that ECMWF HR is 25km grid spacing output at 50km, and LR is 50km output at 100km grid spacing. The delta in grid spacing is therefore 2, and the output are regridded to coarser resolution. This may impact the results.

We added a new Fig 12 showing the correlation between difference and delta.

L. 300-301: Demory et al have revised the manuscript with a focus on EUR-11.

Thanks for pointing that out we change to: "In a similar study Demory et al. (2020) compares PRIMAVERA models with CORDEX LR and CORDEX HR."

L. 306: give extremes that are heavier and more frequent -> simulate more intense and more frequent heavy precipitation. I would avoid the term 'extremes' with such low-resolution models, and refer instead to 'heavy' or 'intense'.

We change to: "They conclude that high resolution models systematically give intense precipitation that is heavier and more frequent."

L. 308: overestimation compared to E-OBS

Changed as suggested

L. 315: CMIP6 and CMIP5

Changed as suggested

L. 318-319: this is probably particularly the case for JJA (as shown in fig 3), but for this conclusion it would be good to see DJF and JJA for fig 6-9.

Information about DJF and JJA for Figs 6-9 are now included in text or in the supplementary.

L. 320: not only. E-OBS is not based on the full network of rain gauges over some other countries, such as France.

We added: "E-OBS is not based on the full network of rain gauges in all countries, which could also lead to undercatch."

L. 332: scale -> grid

Changed as suggested

L. 336: will have -> has

Changed as suggested

L. 340-341: yes but PRIMAVERA tends to be drier than CORDEX in all seasons.

We changed to: "…furthermore GCMs and RCMs of comparable resolution simulate comparable precipitation climates, even though PRIMAVERA is often drier than CORDEX."

L. 343: to -> too

Changed as suggested

L. 344: agree -> agrees

Changed as suggested

L. 345: the quantification can be done if performed on common grids

For the individual models it is possible on common grids, which we now do. On the ensemble level it's more difficult. By difficult we mean that it's not so obvious how resolution influences the ensemble mean because the actual model members used impact the ensemble mean more than the resolution of the members.

L. 348-349: this can depend on seasons

Yes, we addedd the following to the end of the sentence: "especially for heavy precipitation and particularly in summer."

L. 350-351: this needs to be rephrased, as Demory et al have evaluated CMIP5,CORDEX LR/HR and PRIMAVERA HR

We changed to: "The results presented here are in line with previous similar studies using different methods (Demory et al., 2020; Iles et al., 2020) To these studies details are added about the spread between resolutions and between models."

Proper acknowledgement needs to be given to the PRIMAVERA, CORDEX, and CMIP modelling groups

Changed as suggested

There are several typos in the text, please check carefully (e.g. L.8: effects -> affects; L. 20: in depending -> depends; L. 180: region -> regional (and remove comma afterwards); L. 218: were -> where; many others)

These and others are corrected. We apologise for the lack of proof reading, as a reviewer it's annoying to have to correct typos.

L. 139 and 141: below/above c: are these typos?

We removed the "c" for circa as it only confuses.

References:

*Ban N., Schmidli, J., and Schär, C.: Heavy precipitation in a changing climate: Does short-term summer precipitation increase faster?, Geophys. Res. Lett., 42, 1165–1172, https://doi.org/10.1002/2014GL062588, 2015.

*Boé, J., Somot, S., Corre, L., and Nabat, P.: Large differences in Summer climate change over Europe as projected by global and regional climate models: causes and consequences, Clim. Dynam., 54, 2981–3002, https://doi.org/10.1007/s00382-020-05153-1, 2020.

*Delworth TL, Rosati A, Anderson W, Adcroft AJ, Balaji V, Benson R, Dixon K, GriffiesSM, Lee HC, Pacanowski RC, Vecchi GA, Wittenberg AT, Zeng F, Zhang R (2012)Simulated climate and climate change in the GFDL CM2.5 high-resolution coupledclimate model. J Clim 25:2755–2781. doi:10.1175/JCLI-D-11-00316.1

Fernández, J., Frías, M. D., Cabos, W. D., Cofiño, A. S., Domínguez, M., Fita, L.,Gaertner, M. A., García-Díez, M., Gutiérrez, J. M., Jiménez-Guerrero, P., Liguori, G.,Montávez, J. P., Romera, R., and Sánchez, E.: Consistency of climate change projections from multiple global and regional model intercomparison projects, Clim. Dynam.,52, 1139-1156, https://doi.org/10.1007/s00382-018-4181-8, 2019.

*Gutiérrez, C., Somot, S., Nabat, P., Mallet, M., Corre, L., van Meijgaard, E., Perpiñán,O., and Gaertner, M. A.: Future evolution of surface solar radiation and photovoltaic potential in Europe: investigating the role of aerosols, Environ. Res. Lett.,15, 034035, https://doi.org/10.1088/1748-9326/ab6666, 2020.

Hall,A.:Projectingregionalchange,Science,346,1460–1462,https://doi.org/10.1126/science.aaa0629, 2014.

*Kinter III JL, Cash B, Achuthavarier D, Adams J, Altshuler E, Dirmeyer P, Doty B,Huang B, Jin EK, Marx L, Manganello J, Stan C, Wakefield T, Palmer T, HamrudM, Jung T, Miller M, Towers P, Wedi N, Satoh M, Tomita H, Kodama C, Nasuno T,Oouchi K, Yamada Y, Taniguchi H, Andrews P, Baer T, Ezell M, Halloy C, John D,Loftis B, Mohr R, Wong K (2013) Revolutionizing climate modeling with project Athena:a multi- institutional, international collaboration. Bull Am Meteorol Soc 94:231–245.doi:10.1175/BAMS-D-11-00043.1

Klaver R, Haarsma R,Vidale PL, Hazeleger W. Effective resolution inhigh resolutionglobal atmospheric models forclimate studies. Atmos Sci Lett. 2020;21:e952.Na, Y., Fu, Q., & Kodama, C. (2020).Precipitation probability and itsfuture changes from a global cloudâˇAˇRresolving model and CMIP6 simula-tions.Journal of Geophysical Research:Atmospheres,125, e2019JD031926.https://doi.org/10.1029/2019JD031926

Prein, A. F., Gobiet, A., Truhetz, H., Keuler, K., Goergen, K., Teichmann, C., FoxMaule, C., van Meijgaard, E., Déqué, M., Nikulin, G., Vautard, R., Colette, A.,Kjellström, E., and Jacob, D.: Precipitation in the EUROâˇAˇRCORDEX 0.11◦and0.44◦simulations: high resolution, high benefits?, Clim.Dynam., 46, 383-412,https://doi.org/10.1007/s00382-015-2589-y, 2016.

Skamarock, W.C., 2004. Evaluat-ing mesoscale NWP models using kinetic energy spectra. Monthly weather review,132(12), pp.3019-3032.

*Roberts, M. J., Vidale, P. L., Senior, C., Hewitt, H. T., Bates, C., Berthou, S., Chang,P., Christensen, H. M., Danilov, S., Demory, M.-E., Griffies, S. M., Haarsma, R., Jung,T., Martin, G., Minobe, S., Ringler, T., Satoh, M., Schiemann, R., Scoccimarro, E., Stephens, G., and Wehner, M. F.: The Benefits of Global High Resolution for ClimateSimulation: Process

Understanding and the Enabling of Stakeholder Decisions at theRegional Scale, B. Am. Meteorol. Soc., 99, 2341–2359, https://doi.org/10.1175/BAMS-D-15-00320.1, 2018.

Torma, C., Giorgi, F., and Coppola, E.: Added value of regional climate modelingover areas characterized by complex terrainâ˘A˘TPrecipitation over the Alps, J. Geophys.Res. - Atmos., 120, 3957–3972, https://doi.org/10.1002/2014JD022781, 2015.

Vautard, R., Kadygrov, N., Iles, C., Boberg, F., Buonomo, E., Coppola, E., Bülow, K.,Corre, L., van Meijgaard, E., Nogherotto, R., et al.: Assessment of the large EURO-CORDEX regional climate simulation ensemble, J. Geophys. Res. – Atmos., in press.

*Vergara-Temprado, J., Ban, N., Panosetti, D., Schlemmer, L., and Schär, C. (2019).Climate models permit convection at much coarser resolutions than previously consid-ered. J. Clim., JCLI-D-19-0286.1. doi:10.1175/JCLI-D-19- 0286.1.

---

## Author Comment (AC5) · 24 Nov 2020

The paper "The importance of model resolution on simulated precipitation in Europe –from global to regional model" by Strandberg and Lind assesses the ability of a large set of climate models in simulating precipitation (particularly extremes) in European subregions. The authors find that models with coarse grid spacings underestimate the amount and frequency of extreme precipitation but that the variability between models can be larger than the sensitivity to grid spacing. The novel contribution of this study is the inclusion of global climate model data in their analysis since very similar and more detailed analyses have been done with regional models over Europe. I have two major concerns with this manuscript.

We thank the reviewer for making the effort for reviewing our paper and for all comments. Responses follow below in red. In the markup version of the revised manuscript substantial changes are also marked in red.

First, it does not account for the spatial dependence of extreme precipitation. I argue that the authors can obtain the same results by first aggregating E-Obs observations to a coarser grid and then comparing the aggregated extreme precipitation with the original E-Obs data. They would also see that the coarser version of E-Obs "underestimates" extreme frequency and magnitude. Coarse-resolution models should not reproduce the magnitude of extreme events on local scales since they model aggregated rainfall over large areas (e.g., 100x100 km).

The main objective of the paper is not really to focus in precipitation extremes but rather the full distributions (which includes aspects of extremes). We are aware that extremes may not always be well represented in observations, depending on multiple factors including the spatial and temporal character of such events, and we try to acknowledge these weaknesses in the observations in the discussions of the results. As you say the model grid resolution sets limits to what the model can actually resolve but we argue that it is still important to show to what extent different models, from GCMs to RCMs, exhibit similarities and differences in the full precipitation distributions for different regions and seasons.

My second concern is the use of E-Obs for this analysis. E-Obs has very low station density over large parts of Europe and heavily underestimates extreme precipitation. There are other observational datasets available that are far more appropriate for the presented analysis. More details on these comments including relevant literature is provided below.

It is true that E-OBS is inherently associated with uncertainties and the quality is highly dependent on the underlying station density as you say. We intended here to keep the model-observation comparison consistent for all sub-regions by using the same observational data set and hence constrained the comparison to E-OBS solely. We have included a separate section (Sec 2.2) with a discussion of observations and related uncertainties. Furthermore, to highlight the importance of high-quality data sets, we have included in one of the ASoP analyses a regional high-resolution data set (Nordic Gridded Data set, NGCD) that covers the Scandinavian region (see Fig. 3 in Supplementary material). It is seen that NGCD has higher contributions for both low and high precipitation intensities, providing more confidence in especially the RCMs (at least over this region).

General Comments:

1. I have major concerns with your approach to compare extreme precipitation. Extreme precipitation is strongly scale dependent and largest on point scales (e.g. measured by precipitation gauges) and decreases on larger spatial-scales. E-OBS for example has way weaker extreme precipitation than other regional datasets in Europe that feature higher resolution and a higher station density (e.g. Prein and Gobiet 2017). If you compare extreme precipitation on the model native grid, you mix the model ability in simulating extreme precipitation with the spatial scale on which the model simulates extremes. E.g., extreme precipitation in a 100 km grid spacing model should not match observed extreme precipitation on a 25 km grid. In this case the only way to do a fair comparison is to aggregate the 25 km grid observations to the 100km model grid. This aggregation does not introduce large biases such as you state for interpolation (in Line 127-128).

We have now also included analyses where all data are regridded to a 0.5°×0.5° and a 2°×2° grid. This makes it possible for us to separate the effect of model physics from the effect of just having more data points.

2. E-Obs should be used with care for extreme precipitation (Haylock et al. 2008). There are other/regional datasets in Europe that are much better suited for the assessment of extreme precipitation (see Prein and Gobiet et al. 2017).

As mentioned in the response above we have included one other regional data set for the region of Scandinavia (the NGCD data set, see Fig. S3 in Supplementary). However, we would like to emphasize again that extreme precipitation is not the main focus of the study, rather a more holistic approach in the investigation of the model's representation of precipitation over Europe.

3. You are missing to discuss and to refer relevant literature on the ERUO-CORDEX simulations that performed very similar analysis as you present. Kotlarski et al. (2014), Casanueva et al. (2016), and Prein et al. (2016) address similar questions and come to fairly similar conclusions. The novelty of your analysis is that you also include GCM data, which is a valuable contribution but does not change the major conclusions. You should also take a look at Thackeray et al. (2018) who show a highly relevant analysis of model grid spacing and extreme precipitation on a global-scale.

The Introduction has been expanded with a paragraph discussing Demory et al., and Iles et al., as well as other similar studies using CORDEX data:

A few studies have been made investigating how model resolution affects the simulated precipitation in the CORDEX ensembles, comparing 50 km and 12.5 km grid spacing. A clear result is that precipitation generally increases with higher resolution, which sometimes means that the bias increases when precipitation is added to already wet models (Kotlarski et al., 2014; Casanueva et al., 2016); something that is also seen in simulations with global models (e.g. Thackeray et al., 2018). An overall improvement of mean precipitation is not seen the high resolution CORDEX simulations, except for regions with complex topography (Korlarski et al., 2014; Casanueva et al., 2016; Prein et al., 2016). Prein et al. (2016) looked at local precipitation on short time scales. They find that 12.5 km simulations better represent extreme and mean precipitation, also when simulations are aggregated to 50 km. They note, however, that the results are highly dependent on which observations the simulations are

compared with. They also note that improvements are on the ensemble as a whole, and not necessarily for each individual model. In similar studies as the present Iles et al. (2019) and Demory et al. (2020) compare CORDEX simulations with simulations from CMIP5 and Primavera. They see that precipitation increases with resolution so that CMIP5 underestimates precipitation amounts and CORDEX overestimates it, when compared to E-OBS, and that the effect of resolution is largest in complex topography. They also find that Primavera performs similarly to CORDEX when run on the same resolution, which is interesting regarding that the Primavera models are developed for low resolution. Iles et al. (2019) also find considerable inter-model differences meaning that improvements are seen on the ensemble level rather that for individual models.

4.   Please be careful with the use of model resolution. In most cases you refer to model grid spacing.  Model resolution depends on the numeric diffusion in the model and models with the same grid spacing can have different resolutions. The effective resolution of a model is typically 4-8 times its grid spacing (e.g., Skamarock 2004).

Thanks for reminding us about this. We tried to straighten up the terminology so that we use "grid spacing" when talking about distances in km and "resolution" in more general statements, like "comparing high and low resolution models".

5. There are many typos and grammar errors in the document. Please consider using a proofreader before resubmitting the document.

Typos are corrected. We apologise for the lack of proof reading, as a reviewer it's annoying to have to correct typos.

Literature:

Casanueva, A., Kotlarski, S., Herrera, S., Fernández, J., Gutiérrez, J.M.,Boberg, F., Colette, A., Christensen, O.B., Goergen, K., Jacob, D. and Keuler, K., 2016.Daily precipitation statistics in a EURO-CORDEX RCM ensemble: added value of rawand  bias-corrected  high-resolution  simulations.   Climate  dynamics,  47(3-4),  pp.719-737.

Haylock, M.R., Hofstra, N., Klein Tank, A.M.G., Klok, E.J., Jones, P.D. and New, M.,2008.  A European daily highâ˘A˘Rresolution gridded data set of surface temperatureand  precipitation for  1950–2006.   Journal  of  Geophysical  Research:  Atmospheres,113(D20).

Kotlarski, S., Keuler, K., Christensen, O.B., Colette, A., Déqué, M., Gobiet, A., Goer-gen, K., Jacob, D., Lüthi, D., Van Meijgaard, E. and Nikulin, G., 2014. Regional climatemodeling on European scales: a joint standard evaluation of the EURO-CORDEX RCMensemble. Geoscientific Model Development, 7, pp.1297-1333.

Prein, A.F. and Gobiet, A., 2017. Impacts of uncertainties in European gridded precip-itation observations on regional climate analysis. International Journal of Climatology,37(1), pp.305-327.

Prein,  A.F.,  Gobiet,  A.,  Truhetz,  H.,  Keuler,  K.,  Goergen,  K.,  Teichmann,  C.,  Maule,C.F., Van Meijgaard, E., Déqué, M., Nikulin, G. and Vautard, R., 2016.  Precipitationin

the EURO-CORDEX 0.11 and 0.44â´LŸ simulations: high resolution, high benefits?.Climate dynamics, 46(1-2), pp.383-412.

Skamarock, W.C., 2004. Evaluating mesoscale NWP models using kinetic energyspectra. Monthly weather review, 132(12), pp.3019-3032.

Thackeray, C.W., DeAngelis, A.M., Hall, A., Swain, D.L. and Qu, X., 2018. On the connection between global hydrologic sensitivity and regional wet extremes. GeophysicalResearch Letters, 45(20), pp.11-343.

---

## Author Response (AR2)

Review comments for "The importance of model resolution on simulated precipitation in Europe – from global to regional model" by G. Strandberg and P. Lind

I would like to thank the authors for taking into account my comments and responding accordingly. The paper is now much clearer, and the results more robust now that the analyses are also performed on a common grid.

I list below a few additional minor comments. In addition to revising the clarity of the entire manuscript, I would particularly recommend to revise the abstract carefully, clarify its sentences (and correct the typos) so it summarises the findings of this study clearly. I wrote a few suggestions below. I also still found a few typos in the text.

Once again, thanks for constructive comments

L. 12: remove 'weather and', as NWP models are not the scope of this paper

Changed as suggested.

L. 13: 'to represent it': what does 'it' refer to? Precipitation processes, or the changes in precipitation? I would suggest to clarify the entire sentence.

"it" means here precipitation generally. "it" is replaced by "precipitation" and the whole sentence is reformulated to: "Still, due to the complexity of precipitation processes and their large variability in time and space, weather and climate models struggle to represent precipitation accurately."

L. 14: 'in a range of' -> replace by 'in available'

Changed as suggested.

L. 15-17: again here, the word resolution is used when it actually refers to grid spacing. Please specify at which latitude the grid spacing is given.

This seems to have slipped through by mistake. "Resolution" is changed to "grid spacing" and the latitude is specified. This also means that low/high (resolution) is changed to sparse/dense (grid spacing). The sentence now reads: "The ensembles used are: Global climate models (GCMs) from CMIP5 and CMIP6 (~100-300 km horizontal grid spacing at mid-latitudes), GCMs from the PRIMAVERA project at sparse (~80-160 km) and dense (~25-50 km) grid spacing and CORDEX regional climate models (RCMs) at sparse (~50 km) and dense (~12.5 km) grid spacing."

L. 23-24: 'where the largest contribution... and lowest': Does that refer to the differences? I would suggest to clarify.

The sentence is rephrased to: "Overall, in all seasons and regions the largest differences between resolutions are seen for moderate and high precipitation rates, where the largest precipitation rates are seen in the RCMs with highest resolution (i.e. CORDEX 12.5 km) and smallest in the CMIP GCMs."

L. 38: actually, according to Fig 11 and Fig 12, this is also true for PRIMAVERA, but it depends on the index considered. I would suggest to make the sentence more specific to the results shown in the paper.

Not sure what is meant here, both PRIMAVERA and CORDEX is mentioned in this context. Nevertheless, a sentence is added to make it more specific: "For indices describing precipitation days and heavy precipitation (RR1, R20mm, SDII, RX1day) the difference

between two models can be twice as large as the difference between two resolutions, in both the PRIMAVERA and CORDEX ensembles."

L. 39: replace 'higher' by 'increasing'

Changed as suggested.

L. 40: remove 'most often'

Changed as suggested

L. 46: remove 'similar'

Changed as suggested

L. 43-46: it depends on seasons. I would suggest to clarify the whole paragraph based on the results shown.

See answer to comment below

L 46: 'as different CORDEX RCMs driven by the same GCM may give different results' -> they do give different results, but again 1 RCM driven by different GCMs also do give different results (as shown in Fig 11), so do the differences between the RCM and the driving GCM really depend more on the RCM? From Fig 11, it doesn't seem to be the case, both play a role, with more or less impact depending on the regions and seasons. See also results from Vautard et al, 2020, now in press:

Vautard, R., Kadygrov, N., Iles, C., Boberg, F., Buonomo, E., Bülow, K., et al. (2020). Evaluation of the large EURO-CORDEX regional climate model ensemble. Journal of Geophysical Research: Atmospheres, 125, e2019JD032344. Accepted Author Manuscript. https://doi.org/10.1029/2019JD032344

To be clear, we are not saying the simulated climate depends solely on the RCM, we do write: "The result of an RCM simulation depends on the driving GCM". Our point is rather that the difference between the RCM and the driving GCM is not only a result of the higher resolution itself, but also a result of the physics in the RCM. Since this paper is about grid spacings it could easily happen that the reader gets the impression that the change in grid spacing changes everything. If that were the case, then the choice of RCM would be less important; but we do indeed see that the choice of RCM do have an effect. This is not disapproved by the fact that one RCM with several GCMs will also give different results. Since our formulation obviously led to some confusion, and since a proper explanation, like the one above, would be too long for an abstract we removed this sentence.

We added a reference to Vautard et al., 2020 to the Discussion.

L. 47: remove that sentence, as the other studies also look at the spread. The strength of this study is that it looks at many other indices, and includes CMIP6, as well as PRIMAVERA LR, so it adds information on CMIP6 versus CMIP5, and the impact of increasing resolution solely in an ensemble of GCMs.

Changed as suggested.

L. 99: Note that Demory et al do not only compare with E-OBS but also with high resolution observations based on high density network. The published version is:

Demory, M.-E., Berthou, S., Fernández, J., Sørland, S. L., Brogli, R., Roberts, M. J., Beyerle, U., Seddon, J., Haarsma, R., Schär, C., Buonomo, E., Christensen, O. B., Ciarloʼ, J. M., Fealy, R., Nikulin, G., Peano, D., Putrasahan, D., Roberts, C. D., Senan, R., Steger, C.,

Teichmann, C., and Vautard, R.: European daily precipitation according to EURO-CORDEX regional climate models (RCMs) and high-resolution global climate models (GCMs) from the High-Resolution Model Intercomparison Project (HighResMIP), Geosci. Model Dev., 13, 5485–5506, https://doi.org/10.5194/gmd-13-5485-2020, 2020.

We make this clearer by this (new words in italic):

"The results show that precipitation increases with resolution and that, when compared to *a mixture of* E-OBS *and high spatial-resolution gridded national datasets*"

L. 153: add 50 km mid-latitude grid spacing

Changed as suggested.

L. 930: 'is given in the same format as'

Changed as suggested.

Table 2: institue -> institute; Euro-CORDEX-> EURO-CORDEX

Changed as suggested.

Fig. 2 and 3: specify in the caption that it is now the median which is shown.

Changed as suggested.

Pay also attention to the use of the word 'different/difference', it is used many times in single sentences (e.g. L. 18, L. 554, many others) and makes the sentences hard to read.

We see what you mean. Some of these "different/difference" are removed when they are not really needed and just add confusion; some others are replaced by other words when possible.